

# Glacier equilibrium line altitude variations during the "Little Ice Age" in the Mediterranean Andes (30° - 37°S)

Álvaro González-Reyes[1], Claudio Bravo[2], Mathias Vuille[3], Martin Jacques-Coper[4], Maisa Rojas[5], Esteban Sagredo[6], and James McPhee[7,8]

[1]Laboratorio de Dendrocronología y Cambio Global, Facultad de Ciencias Forestales y Recursos Naturales, Universidad Austral de Chile, Valdivia, Chile
[2]School of Geography, University of Leeds, Leeds, UK
[3]Department of Atmospheric and Environmental Sciences, University at Albany, State University of New York, Albany, NY, USA
[4]Departamento de Geofísica and Center for Climate and Resilience Research (CR)2, Universidad de Concepción, Concepción, Chile
[5]Departamento de Geofísica and Center for Climate and Resilience Research (CR)2, FCFM, Universidad de Chile, Santiago, Chile
[6]Departamento de Geografía, Pontificia Universidad Católica de Chile, Santiago, Chile
[7]Departamento Ingeníeria Civil, FCFM, Universidad de Chile, Santiago, Chile
[8]Advanced Mining Technology Center AMTC, FCFM, Universidad de Chile, Santiago, Chile

**Correspondence:** Álvaro González-Reyes (gonzalezreyesalvaro@gmail.com)

**Abstract.** The "Little Ice Age" (LIA; 1500 – 1850 Common Era (CE)), has long been recognized as the last period when mountain glaciers in many regions of the Northern Hemisphere (NH) recorded extensive growth intervals in terms of their ice mass and frontal position. The knowledge about this relevant paleoclimatic interval is vast in mountainous regions such as the

Alps and Rocky Mountains in North America. However, in extra-tropical Andean sub-regions such as the Mediterranean Andes of Chile and Argentina (MA; 30º - 37ºS), the LIA has been poorly documented. Paradoxically, the few climate reconstructions performed in the MA based on lake sediments and tree rings do not show clear evidence of a LIA climate anomaly as observed in the NH. In addition, recent studies have demonstrated temporal differences between mean air temperature variations across the last millennium between both hemispheres. This motivates our hypothesis that the LIA period was not associated with

a significant climate perturbation in the MA region. Considering this background, we performed an experiment using daily climatic variables from three Global Climate Models (GCMs) to force a novel glaciological model. In this way, we simulated temporal variations of the glacier equilibrium-line altitude (ELA) to evaluate the glacier response during the period 1500 – 1848 CE. Overall, each GCM shows temporal changes in annual ELA, with anomalously low elevations during 1640 – 1670 and 1800 – 1848 CE. An interval with high ELA values was identified during 1550 – 1575 CE. The spectral properties of

the mean annual ELA in each GCM present significant periodicities between 2 – 7 years, and also significant decadal to multi-decadal signals. In addition, significant and coherent cycles at interannual to multi-decadal scales were detected between modeled mean annual ELAs and the first EOF1 extracted from Sea Surface Temperature (SST) within the El Niño 3.4 of each GCM. Finally, significant Pearson correlation coefficients were obtained between the mean annual ELA and Pacific SST



on interannual to multi-decadal timescales. According to our findings, we propose that Pacific SST variability was the main
20    modulator of temporal changes of the ELA in the MA region of South America during 1500 – 1848 CE.

# 1    Introduction

The term "Little Ice Age" (LIA) usually refers to climatic anomalies over the Northern Hemisphere between the 15th and
mid-19th centuries. During this interval, a large number of mountain glaciers in the Northern Hemisphere evidenced frontal
25    advances (Solomina et al., 2008). The LIA has been well documented in the Northern Hemisphere by means of climatic proxies
and historical documents. Some examples from tree rings in North America show reductions in summer temperatures since
1400 CE (Luckman and Wilson, 2005). In the case of Europe, studies carried out by Nussbaumer et al. (2011) identified histor-
ical glacier fluctuations of Jostedalsbreen and Folgefonna glaciers (Southern Norway) using pictorial and historical evidence.
The climate variability, and the magnitude and amplitude of changes that control glacier mass balance during this period are
well understood in many regions of the Northern Hemisphere (Davis, 1988; Grove, 1988; Solomina et al., 2008, 2015). How-
ever, in extra-tropical regions of the Southern Hemisphere, such as the Mediterranean Andes region of Chile and Argentina
(MA; 30º – 37ºS, Figure 1), the LIA evidence and glacier response to climatic forcings remains poorly analyzed. Only a small
number of studies has been developed in South America to explore possible evidence for a regional climate anomaly during the
LIA. These studies have been located in the Tropical Andes of Peru (Solomina et al., 2007; Bird et al., 2011), Bolivia (Rabatel
et al., 2008) and Ecuador (Ledru et al., 2013). The approach has focused on frontal length variations (Masiokas et al., 2009;
Araneda et al., 2009), peat dating (Espizua and Pitte, 2009) and climatic proxies such as lichenometry (Rabatel et al., 2008;
Ruiz et al., 2012). However, while glacier mass balance modeling and estimation of equilibrium-line altitude (ELA) have been
used in the Tropical Andes (e.g., Jomelli et al., 2011), up to date, they have not been applied in the MA region. On the other
hand, several paleoclimate reconstructions have been developed in the MA. For instance, a precipitation reconstruction based
on tree rings spans the last millennium (LeQuesne et al., 2006). Nevertheless, neither this reconstruction nor a reconstruction of
Santiago de Chile precipitation performed by LeQuesne et al. (2009) show clear evidence of decreased or increased in rainfall
during 1500 - 1850 CE. Aside from regional climate reconstructions, large-scale estimates of inter-hemispheric differences
in mean air temperature developed for the last millennium by Neukom et al. (2014) are equally important to understand the
regional influence of the LIA climate anomaly in the MA. This information is relevant in order to provide informed hypotheses,
such as whether the LIA left a significant imprint in the mid-latitudes of the Southern Hemisphere, or whether other, more local
factors, exerted a dominant control over glacier mass balance and ELA within this period in regions such as the MA.

The ELA is a climate-sensitive parameter and it is theoretically defined as the location on a glacier where annual accumu-
lation is balanced by annual ablation, i.e., net surface mass balance equals zero (Rupper and Roe, 2008; Cogley et al., 2011).
Therefore, the ELA can be estimated by fitting a curve to data representing surface mass balance as a function of elevation



(Cogley et al., 2011). The ELA is a good indicator of the general conditions of the glacier response at regional scale, as glacier

mass balances are well correlated at distances of up to 500 km (Letreguilly and Reynaud, 1989). Because the ELA is primarily

forced by mean air temperature and precipitation (Porter, 1975; Rupper and Roe, 2008; Sagredo et al., 2014), ELA estimates

are commonly used in palaeoclimatic reconstructions.

Given the location of the MA along the west coast of South America, we expect that Pacific sea surface temperature (SST)

variability exerts a strong control on the ELA behavior of Mediterranean Andean glaciers in Chile and Argentina during this

late-Holocene interval. The aim of this study is thus to use a novel glacier mass balance modeling procedure to estimate the

ELA of these glaciers, and to examine the possible role of Pacific (SST) variability on their surface mass balance during 1500

- 1850 CE.

## 2 Data and Methodology

### 2.1 Experimental setup

We use a novel glacier mass balance model forced with Coupled Model Intercomparison Project – Phase 5 (CMIP5) model

data for the period 1500 - 1848 CE. In the next sections we explain the details of the glaciological model and its set-up. The

general approach consists of estimating the theoretical surface glacier mass balance (MB) in each model grid cell as a function

of elevation, provided by climate variables from three General Circulation Models (GCMs). We used the ELA as a general

indicator of glacier behavior as we are not considering individual glaciers and their specific responses to climatic variations.

For each model grid point we obtained the MB as a function of altitude. From this MB profile we thus obtained a gridded ELA

estimate for each year. We focus on the regional response of ELA in the MA region (Figure 1).

#### 2.1.1 Glacier mass balance model

A simple glacier mass balance model was used in order to estimate the annual ELA variations between 1500 - 1849 CE.

Previous paleoclimate studies around the globe have used a degree-day model forced by data from GCMs to obtain past

glacier variations (Hostetler and Clark, 2000). In addition, several studies focused on understanding ELA changes across the

Holocene period. For instance, Rupper et al. (2009) applied a similar model to the region of Central Asia, using data from the

NCEP/NCAR reanalysis for the present and data from GCMs of phase 1 of the Paleoclimate Model Intercomparison Project

(PMIP). In the Southern Hemisphere, Bravo et al. (2015) applied this kind of model forced by PMIP2 data to compare past

ELA conditions between the Southern Alps (New Zealand) and Patagonia during the Mid-Holocene (6000 BP).

The glacier mass balance model used in this study was developed by Anderson and Mackintosh (2006) and applied to Franz

Josef Glacier (Southern Alps). A detailed description of this model can be found there. In general, the model calculates the

mass balance gradient for any specific location, based on daily data of temperature and precipitation as a function of elevation.

In glaciological terms, the degree-day method shows better performance in latitudes where there are no other losses of ice mass



by external forcings other than temperature (e.g. Patagonia). In extra-tropical regions such as the MA, studies carried out by
Pellicciotti et al. (2008) and Ayala et al. (2017) have demonstrated that solar radiation is intense. For instance, Pellicciotti et al.
(2008), using field measurements confirmed that the incoming solar radiation can reach values close to the solar constant ($\sim$
$1367\ W \cdot m^{-2}$) in the Juncal Norte glacier (32.98°S, 70.11°W). For this reason, we modified the model developed by Ander-
son and Mackintosh (2006). In our case, the melt computation adopts the version of Pellicciotti et al. (2005). Furthermore, we
assume that the present-day solar radiation effect on Andean glaciers located in the MA region has persisted across the last
millennium.

Elevation in our model is defined in each grid point from 100 to 6000 m above sea level (a.s.l.) with increments of 20 m.
We consider this elevation range due to the high-altitude peaks within this region (e.g. Aconcagua, Marmolejo, Mercedario,
Tupungato, among others, all reaching altitudes above 6000 m a.s.l). For each elevation, the mass balance is calculated based
on:

$$\dot{m}(t, z) = \dot{c}(t, z) + \dot{a}(t, z) \tag{1}$$

Where $\dot{m}$ is the mass balance rate, $\dot{c}$ the accumulation rate, and $\dot{a}$ the ablation rate at time t and elevation z. In the glacier
mass balance model, accumulation is defined as the portion of the daily precipitation that falls as snow when the daily average
temperature is below a certain temperature threshold ($T_{crit}$). Radic and Hock (2011) considered an optimal $T_{crit}$ in the range
of 0° to 2°C. In our case, $T_{crit}$ was assumed to be 1°C (Anderson and Mackintosh, 2006). Therefore, water equivalent (w.e.)
accumulation is calculated based on the daily information of mean temperature ($T_{mean}$) and total daily precipitation ($P_d$),
calculated as

$$c(t, z) = \delta_m P_d \begin{cases} \delta_m = 1, & T_{mean} < T_{crit} \\ \delta_m = 0, & T_{mean} \geq T_{crit} \end{cases} \tag{2}$$

Despite the fact that temperature is a good predictor of melting, because incoming shortwave radiation and turbulent heat
fluxes are closely related to air temperature (Oerlemans, 2001; Ohmura, 2001), it is relevant to consider these variables sepa-
rately, particularly in extra-tropical regions, such as the MA. Based on this, the ablation ($a$) following Pellicciotti et al. (2005)
is calculated according to the relationship given by

$$a(t, z) = \begin{cases} TF \cdot T_{mean}(t, z) + SRF \cdot (1 - \alpha(t, z)) \cdot G(t), & T_{mean} > T_{crit} \\ 0, & T_{mean} \leq T_{crit} \end{cases} \tag{3}$$

Where $T_{mean}$ represents the mean air temperature at $t$ time and $z$ elevation. $G(t)$ is the solar radiation at Earth surface
at $t$ time, and $\alpha(t, z)$ is the surface albedo given by fresh snow, snowpack and ice according to elevation z. We used albedo





values of 0.85 for fresh snow, 0.60 for snowpack and 0.30 for ice (Cuffey and Paterson, 2010). In general, albedo is higher for snow and lower for ice. Two empirical coefficients of temperature ($TF$) and short-wave radiation ($SRF$) factors are included.

The $TF$ y $SRF$ factors are based on empirical measurements obtained from the Juncal Norte and San Francisco glacier, both located in the MA region. We considered values of 90 (mm $\cdot h^{-1} \cdot °C$) $\cdot 10^{-4}$ for $TF$, and 100 (m$^2$ mm $\cdot W^{-2} \cdot h^{-1}$) $\cdot 10^{-4}$ for $SRF$. Similar values for $TF$ and $SRF$ factors are reported by Ayala et al. (2017) obtained from different glaciers located in the MA region. We assume both factors as constant values throughout the day and throughout the analyzed period. The solar radiation and albedo factors were parameterized and calculated as described in the Appendix section. As stated in eq. 3, we

calculated ablation only when T$_{mean}$ was positive (T$_{mean} > 0$), otherwise we considered ablation equal to 0.

## 2.2 Comparison between observed and modeled ELA in MA

Because this model has not been previously applied in this region, we compared the estimated against observed ELA at the Universidad Glacier, located in the MA (34° 40S; 70° 20W). There are scarce measurements of surface mass balance and ELA data in the region. For instance, a surface mass balance record of Echaurren Norte glacier exists, which spans the period 1976

- 2013 (Masiokas et al., 2016). However, there is no ELA information available for this glacier. In the case of Universidad glacier, recent measurements of the ELA indicate a value of 3497 m a.s.l. for the 2000 year (Carrasco et al., 2005); between 3500 and 3700 m a.s.l for the 2009–2010 hydrologic year (Bravo et al., 2017). In addition, studies carry out by Kinnard et al. (2018) report values of 3478 m and 4233 m for the 2012 - 2013 and 2013 - 2014 hydrologic year, respectively.

## 2.3 Model inputs: CMIP5

We used daily climate data from three GCMs based on past1000 experiment simulations (runs r1i1p1) of the CMIP5 initiative (Table 1). For each GCM, we extracted the daily mean, minimum and maximum air temperature and precipitation for the period 1500 - 1849 CE. We used a hydrological year, considering April to March months. For this reason, our analysis period covered the 1500 - 1848 CE period. From each GCM, we only considered climate information from grid cells that span across 70°W, in order to extract only information over the Andes. This restriction was applied due to the coarse resolution of the GCMs used to

estimate the ELA in mountain areas of the MA. Additionally, topographic information from the respective GCM and specific grid point information were used to compute temperature and precipitation lapse rates. In order to evaluate the capability of GCMs to reproduce the annual climatology of the MA region, we compare monthly precipitation and mean air temperatures from GCMs based on Historical experiment simulations with data from the El Yeso meteorological station (YESO; 33°40′S; 70°05′W; 2475 m, no missing data during this period). We compare both datasets over the 1979 – 2010 years.

Mean air temperature data were calculated for different elevations using a standard and constant lapse rate of -6.5°C $\cdot$ km$^{-1}$. Due to the scarce number of studies about glacier-climate interactions in this part of the Andes, for minimum and maximum temperatures we used a constant lapse rate value of -5.5°C $\cdot$ km$^{-1}$ following studies carried out in the Tropical Andes by Córdova et al. (2016). In the case of precipitation, and given that the distribution of precipitation in mountainous regions is difficult to predict even under present-day conditions (Rowan et al., 2014), we use a constant rate of 0.02 mm $\cdot$ m$^{-1}$ in

order to facilitate the computation of mass balance modelling and ELA estimation. Based on each GCM, we calculated daily



ELA in each specific grid spanning the period 1500 - 1848 CE. Seasonal values for austral winter and summer ELA intervals were estimated as well. Austral winter spans the months between May and August, while summer spans the months between December and March. The annual and seasonal ELA estimated for each grid cell was averaged over the MA domain in each GCM in order to construct the annual mean and seasonal ELA. Additionally, we calculate the composite annual ELA, which

corresponds to the mean of the three GCMs. We define this as the Regional ELA. The purpose of this study was to identify periods of glacier advance and retreat based on ELA behavior; if the ELA is lowered, then the glaciers advance and if it rises, the glaciers retreat (Rupper and Roe, 2008). We used as reference the long – term mean elevation during 1500 - 1848 CE.

## 2.4 ELA - ENSO relationship

In order to test our working hypothesis of a strong SST influence on the regional ELA, we extracted the ENSO signal of

each GCM using Empirical Orthogonal Functions (EOF) from daily SST data. We used the first principal component (PC1) or first leading EOF calculated within the Niño 3.4 region in the tropical Pacific domain (5°N - 5°S, 170°W - 120°W). We used EOF's to estimate spatial and temporal variations of SST anomalies within the Niño 3.4 region between April and March during 1500 - 1848 CE. The EOF analysis was computed with the R statistical software using the "Sinkr" package (Taylor, 2017). A complete guide to EOF analysis can be found in von Storch and Zwiers (2001).

## 155 2.5 Spectral properties

To identify the main periodicities in the variations of the regional ELA in each GCM, we performed a spectral analysis using the Multi Taper Method spectrum MTM (Mann and Lees, 1996). Furthermore, the continuous and coherence wavelet transform were used to detect main periodicities and coherence within the time series. An exhaustive guide to MTM and SSA spectral analyses is found in Ghil et al. (2002). The Continuous Wavelet Transform (CWT) provides information of the main

periodicities in a time-frequency domain, while cross wavelet (XWT) and coherence wavelet analysis allows the detection of common and high spectral power signals, and periods where two-time series present spectral coherence, given a specific statistical confidence level in a specific time t. In all cases, we used the "morlet" mother wavelet to compute both wavelet spectral analyses. More information about wavelet computation and theory is found in Torrence and Compo (1998). The computation of wavelets was performed with the R statistical software using the "biwavelet" package (Gouhier et al., 2017). For contrasting

relationships at multi-decadal scales, we used a LOESS filter of 21 years applied over ELA and SST Niño 3.4 time series in each GCM. Finally, we calculated fields of Pearson correlation coefficients for 1500-1848 CE between each SST dataset and the regional ELA time series. For the latter analysis, we focused on two relevant time resolutions: the interannual and the decadal time scales.





## 3   Results

### 3.1   Glaciological model

Our glaciological model seems to reproduce the ELA of the MA region, however we observed an overestimation of observed ELA values (Figure A1). We need to keep in mind the potential sources of error in the mass balance model. For instance, the use of a constant and linear precipitation lapse rate ($0.02 \ mm \cdot m^{-1}$) and temperature lapse rate ($-6.5 \ °C \cdot km^{-1}$) and solar radiation parameterization may not be the most appropriate approximation to represent the real conditions of some areas. To quantify the uncertainty, we computed the mean annual ELA for Universidad glacier located in the MA region (34° 40S; 70° 20W; Figure 1). Daily gridded precipitation and mean, minimum and maximum temperatures with 0.005º x 0.005º spatial resolution from the Center of Climate and Resilience Research (CR)2 were used to compute ELA values for the period 1979 – 2015, using the April – March hydrological year. More information about the dataset can be obtained from CR2met web page http://www.cr2.cl/datos-productos-grillados/. This gridded product is based on ERA-Interim (Dee et al., 2011), MODIS satellite images and measurements from meteorological stations. We extracted 16 grid values from the cell encompassing Universidad glacier (34.3º - 34.5ºS; 70.1º - 70.3ºW). We used the same glaciological parameters reported in the methods section to run our glaciological model over the Universidad glacier domain. Despite the existence of many mountain glaciers across the MA region, there exists only a small number of glaciers with ELA estimates. We compare our results with in situ and satellite-based ELA estimates presented in previous published works. Our result are consistent with ELA values reported by (Carrasco et al., 2005; Bravo et al., 2017; Kinnard et al., 2018) during 2000, 2009, 2012 and 2013, respectively (Figure A1). In all cases the annual observed ELA is within the range of the modeled ELA. However, in terms of absolute values, we found some discrepancies in the ELA identified by Carrasco et al. (2005) and Kinnard et al. (2018) during 2000 and 2012, respectively. Carrasco et al. (2005) report a value of 3497 m.a.s.l. for Universidad glacier in 2000, while we obtained a modelled median ELA equal to 3722 m.a.s.l. On the other hand, Kinnard et al. (2018) report the ELA to be located at 3478 m a.s.l. in 2012, which is inconsistent with the median of 3980 m a.s.l. of our modelled ELA at Universidad glacier. On the other hand, despite to uncertainties of GCMs and based on the ELA equations reported by Carrasco et al. (2005, 2008), the present ELA is located at 4083 m.a.s.l for glaciers between 30° - 37°S, while our regional mean annual ELA during 1500 – 1848 CE was 3745 m.

### 3.2   Climate data input

The climatic data from the three GCMs from past1000 CMIP5 used in the glacier mass balance model to determine the ELA, reproduce the annual climatology from El Yeso meteorological station (YESO) located in the MA at 2475 m (Figure 2) quite well. A similar result is found when comparing YESO and Historical CMIP5 runs. The past1000 runs simulate increased precipitation during the austral winter months (May to August), followed by an extended period of low precipitation rates until March (dry conditions during austral summer). The highest monthly precipitation values are observed in the NCAR-CCSM4 model during austral winter months (May to August; Figure 2a), while the MPI-ESM-P model registers the lowest amount in this season. Precipitation amounts in austral summer (December to March) are higher in MRI-CGCM3 and NCAR-CCSM4



models than at the YESO station. Similar precipitation amounts, however, are observed at YESO and in the MPI-ESM-P model. The comparison of mean air temperature shows a good correspondence between models and YESO during austral winter (May to August), particularly the MRI-CGCM3 and MPI-ESM-P models (Figure 2b). All GCMs exhibit their lowest

monthly temperature in July. During austral summer GCMs show coherent monthly variations with highest values in January and February.

### 3.3 ELA temporal variations

Our glaciological model forced by daily climatic data from three GCMs from CMIP5 past1000 experiments allowed simulating the ELA variations across the 100 to 6000 m a.s.l altitudinal gradient, between 1500 and 1848 CE. In a long-term mean context,

lowest ELA values were found using the MRI-CGCM3 model (3565.2 m; Figure 3a), while highest ELA values have been obtained by NCAR-CCSM4 model (3995.5 m; Figure 3c). In addition, this model presented a higher standard deviation of the annual ELA, compared with the other two models. In a regional context, our results suggest that the annual regional long-term mean glacier ELA was located at 3745 m a.s.l. The minimum annual elevation in the regional ELA was equal to 3380 m a.s.l. in 1817, while the annual maximum elevation was located at 4120 m in 1777. A series of pulses characterized by low and

high ELA values were identified in each simulation and also dominate the regional ELA reconstruction (Figure 3). Periods with low ELA values are identified during 1600 – 1650 CE in all GCMs, and during 1800 – 1848 CE in MRI-CGCM3 and NCAR-CCSM4 (Figure 3a,c). High ELA values are apparent during 1560 – 1580 CE and 1750 – 1770 CE in MPI-ESM-P and NCAR-CCSM4 (Figure 3b,c). In a regional context, an increase of the mean annual regional ELA could be observed during 1550 - 1575 CE, while a decrease of ELA values is identified between 1640 - 1670 and 1800 - 1848 CE (Figure 3d). On the

other hand, if we consider the spatial pattern of mean annual ELA within our domain of analysis, a consistent increase from south to north is evident, similar to results presented by Carrasco et al. (2008) under present climate conditions (Figure A2).

### 3.4 Spectral properties of the regional annual ELA

Spectral properties of the mean annual ELA time series obtained from each GCM reveal significant and common periodicities, mainly between 2 - 7 and 8 - 16 years (Figure 4), suggesting that the mean annual ELA in the MA region was varying

at preferred periodicities on interannual to multi-decadal timescales during 1500 - 1848 CE. In the time-frequency domain, significant and similar spectral signals were detected when applying the CWT analysis on the mean annual ELA from MRI-CGCM3 and MPI-ESM-P models. Interannual variability with periodicities around 2 – 7 and 8 – 16 years shows a cluster during 1650 - 1750 CE (Figure 4b,d). Furthermore, a strong and significant (P < 0.05) signal is identified in the annual ELA from the MPI-ESM-P model with periodicities between 16 - 32 years. After the 1750s, the CWT of the annual ELA from all

GCMs shows significant signals associated with interannual to multi-decadal time scales (high to low frequency cycles). All spectral signals located within the thick black contours of the CWT analysis (see Figure 4) present a statistical significance at P < 0.05.



## 3.5 Relationships between regional annual ELA and ENSO

We compared the mean annual ELA of each GCM with the first principal component obtained from an EOF analysis of SST

from the Niño 3.4 region, considering hydrologic years from April to March (Figure A3). In all models, the first principal
component or EOF1 retained more that 69% of the total variance. Similar results in terms of percentage explained by EOF1
were obtained when averaging over the months from July to June. Relationships in the frequency domain revealed by XWT
analysis show strong and significant joint periodicities around 2 – 7 years, inherent in both the mean annual ELA and EOF1
SST. This result can be observed in all models (Figure 5a,c,f), and throughout the whole period analyzed. In addition, significant

joint periodicities associated with 8 - 32 years were identified mainly between the mean annual ELA and the EOF1 SST of
the MPI-ESM-P model (Figure 5c). Periods when the relationship between the ELA and SST is in-phase and anti-phased can
be deduced from the XWT analysis, and is illustrated by the direction of the arrows. Similar results were identified using
the wavelet coherence analysis, where significant and coherent spectral signals shows a concentration between 2 - 7 years of
periodicity, and at mid- and low frequencies associated with 8 - 32 years (Figure 5b,d,f). In addition, a significant periodicity

between 24 - 32 years is observed after 1650, particularly in the MPI-ESM-P model (Figure 5d), and between 32 – 64 years
since 1700 in the NCAR-CCSM4 model (Figure 5f). In addition to significant spectral coherence between mean annual ELA
and EOF1 SST from th El Niño 3.4 region, we also found significant relationships between the mean annual ELA and SST over
the whole Pacific domain on interannual to multi-decadal time scales. Pearson's field correlations between the mean annual
ELA and mean annual SST from April to March months based on the MRI-CGCM3 model show statistical significance over the

tropical Pacific region at interannual scales (Figure 6a). Similar results can be observed using the MRI-ESM-P model, which
shows a strong tropical Pacific signal (Figure 6c). The spatial correlation field from NCAR-CCSM4 shows strong similarities
with the one from the MRI-CGCM3 model, particularly in the tropical Pacific region (Figure 6e). The same analysis shows
an increase in correlation coefficients, when Pearson's correlations are calculated with 21-year LOESS-filtered time series,
especially when considering the MPI-ESM-P and NCAR-CCSM4 models (Figure 6d and 6f; respectively).

## 4   Discussion


In this study we modeled the temporal variability of the ELA across the Mediterranean Andes region from 1500 to 1848 CE.
This methodological approach is novel, as it is the first study which evaluates the temporal variability of the ELA forced by
GCMs in order to understand the Andean glacier response to climate variability during the period 1500 - 1848 CE. Our results
show that CMIP5 models used in this study reproduce well the modern annual climatology observed in the MA region. How-

ever, in terms to modern ELA we founded discrepancies between past1000 and Historical runs. We observed lower ELA values
using Historical runs between 1850 - 2005 compared to past1000 runs during 1500 - 1848 CE, and using the MPI-CGCM3 and
NCAR-CCSM4 models. In the case of MPI-ESM-P, this GCM has reproduced lower ELA values during 1500 – 1848 compared
to modern times given by Historical runs. On the other hand, historical and past1000 runs shows strong similarities across the
three models used to reproduce precipitation and mean air temperature variations over the MA region. Nonetheless, summer

precipitation is overestimated in the MRI-CGCM3 and NCAR-CCSM4 models when compared with YESO meteorological



station (Figure 2a). Similar results were found when considering mean air temperature, with an overestimation by all GCMs in the summer season (Figure 2b).

The results of our modeled annual mean ELA show that the lowest value is given by the MRI-CGCM3 model, while the highest annual mean ELA value is obtained using the NCAR-CCSM4. The regional long-term ELA mean was estimated at 3745 m a.s.l. This value is lower than the 4083 m a.s.l. estimated under present climate conditions and for the same region by Carrasco et al. (2005, 2008). Despite to this, the ELA magnitude should be interpreted with caution because we have observed an overestimation of ELA values with our model. The time series show an important interannual variability in ELA values for each of the GCMs (Figure 3a,b and c). This variability is also observed in the averaged time series of the regional ELA during 1500 - 1848 CE (Figure 3d). It seems that high ELA values may be related to differences in the monthly precipitation amounts obtained between GCMs, as NCAR-CCSM4 present the highest values. This result agrees with the findings of Masiokas et al. (2016) who estimate a greater glacier mass balance sensitivity to precipitation in central Chile.

Our modeled ELA in the MA region does not show longer intervals with a sustained low/high ELA (associated with positive/negative glacier mass balance), as identified in the Northern Hemisphere during the second half of the last millennium through lake sediments (Bakke et al., 2005), tree rings (Linderholm et al., 2007) and multiple climate proxies (Solomina et al., 2007). In a regional context, our modeled ELA shows only three notable periods where it departed significantly from the long-term mean for a longer period of time. We found that the mean ELA was consistently located at an anomalously high elevation during 1550 - 1575 CE, while during 1640 - 1670 and 1800 - 1848 CE, ELA values were lower than the long-term mean value. During 1550 - 1575 CE, mean air temperature during both winter and summer seasons was similar to the long-term mean. However, precipitation amounts were anomalously high during this period (Figure A4). In addition, spectral properties revealed by XWT and wavelet coherence analyses show significant periodicities around 2 – 7 years during this period (see Figure 5), which may be related to ENSO variability. A study carried out by Montecinos and Aceituno (2003) shows that winter precipitation amounts in the MA region are above average during the warm (El Niño) phase of ENSO. During the period with low ELA values (1640 - 1670 CE), precipitation and temperature do not show any clear departures from the mean (Figure A4). However, winter and summer precipitation increase during 1810 - 1820 CE, with strong mean air temperature reductions observed in both seasons (Figure A4). The combination of high precipitation amounts and lowered mean air temperature could explain the low ELA values in both the MPI-ESM-P and NCAR-CCSM4 models during 1800 - 1848 CE. In addition, after 1815 the regional annual ELAs reached their lowest elevations within the analyzed interval in all models. The strong temperature reduction observed during this period is likely the result of volcanic forcing in the GCMs, in particular associated with the 1815 eruption of Tambora in Indonesia (Self et al., 1984). In addition, recent studies carried out by Wang et al. (2017) using the Bergen Climate Model during 1400 - 1999 CE indicate that ENSO shows a negative-positive-negative response to strong tropical volcanic eruptions, which corresponds to the different stages of volcanic forcing. In our case, we used the Superposed Epoch Analysis (SEA; (Prager and Hoenig, 1989) to contrast our mean annual ELAs with an external forcing such as large volcanic eruptions occurred during the years: 1585, 1601, 1641, 1698, 1783 and 1816, reported by Schneider et al. (2017). We



obtained significantly lower ELA values with positive lags between one to three years (Figure A5). In addition, we identified

the lowest ELA values within the period 1815 – 1820, coincident with a large volcanic eruption (see Figure 3). This result

indicates that large scale eruptions might generate variations in glacier dynamics such as ELA changes in the MA region. On

the other hand, during 1800 – 1848 CE, significant spectral cycles of 8 – 16 years were detected by XWT and wavelet coher-

ence using the MPI-ESM-P and NCAR-CCSM4 models (Figure 5). The spectral signals associated to 2 – 7 years, identified

by each mean annual ELA derived from each GCM was also identified in regional precipitation reconstructions performed by

LeQuesne et al. (2006), and in Santiago de Chile precipitation reconstruction (LeQuesne et al., 2009). Both reconstructions are

based on tree rings as climate proxy.

Using spectral analysis, we identified strong and significant signals between the time series from each annual ELA and the

mean April-March SST within the El Niño 3.4 region, for each GCM. These cycles present periodicities between 2 – 7 years.

However, significant spectral signal at decadal to multi decadal scales were identified as well during 1500 - 1848 CE (Figure 5).

During 1550 - 1580 CE, our modeled ELA shows an increase. In addition, within this interval we found negative values in the

austral winter precipitation, together with air temperature values above the long-term mean during summer (Figure A4), repre-

senting a warm period. MPI-ESM-P and NCAR-CCSM4 in MA mainly exhibit a combination of high mean air temperatures

with an absence of winter precipitation. We observed that mean air temperature has not changed dramatically within 1500

- 1848 CE in MA, in comparison to other relevant climatic periods within the last millennia, such as the Medieval Climate

Anomaly (MCA). Recent studies developed by Rojas et al. (2016) found small temperature changes during this period and

different intervals within 1500 - 1848 CE using the same GCMs of our study. Moreover, paleoclimatic studies such as DeJong

et al. (2013), using varves extracted from the Laguna Chepical (3050 m a.s.l. ; 32°S), indicate a predominance of cool summer

conditions during the period analyzed in this work. In our case, we observe a marked decadal to multi-decadal variability in

mean summer air temperature obtained from GCMs (Figure A4). However, a similar declining pattern in summer temperatures

is observed during 1800 - 1820 CE when comparing the Laguna Chepical reconstruction and the GCMs used in our study. In

addition, reductions of mean air temperature in winter (May to August) coincide with intervals when the regional annual mean

ELA exhibit low elevations (e.g. around 1700s).

Our modeled ELA is characterized by strong decadal to multi-decadal variations during the analyzed period. We obtained

significant relationships between mean annual tropical Pacific SST and mean annual ELA for each GCM after applying a 21-

year LOESS filter (Figure 6). The dominant influence of the Pacific SST variability seems to mask out the LIA signal that has

been reported by many studies in the Northern Hemisphere (e.g., Luckman and Wilson, 2005; Solomina et al., 2015). Still, our

results indicate low ELA values around 1840 (Figure 3), consistent with a maximum advance in 1842 of Glacier Cipreses, lo-

cated in the MA region, as documented by Araneda et al. (2009). This advance has been associated with the LIA, however, our

work suggests that this advance may have been a response to a more short-lived climate anomaly during the period 1800-1848

CE. The regional annual ELA does not indicate a prolonged cooling interval between 1500-1848 CE as it has been identified

in the Northern Hemisphere based on temperature reconstructions using tree rings (Mann et al., 1999; Luckman and Wilson,



2005) and GCMs (Neukom et al., 2014). Our experiment suggests that there was no severe and extended cooling interval within the MA coeval with the Northern hemisphere LIA period. Future studies should focus on unravelling the LIA signal in other regions of the Andes, e.g., Patagonia, where geomorphological evidence suggests glacial advances occurred during this period (Koch and Kilian, 2005; Rodbell et al., 2009; Rivera et al., 2012; Aniya, 2013).



## 5 Conclusions

In this work, we simulated the daily equilibrium line altitude (ELA) representative of glaciers located in the Mediterranean Andes region of South America. Our approach used a novel glaciological model to estimate surface mass balance throughout an altitudinal gradient during 1500 - 1848 CE. We focused on the general response of the regional annual ELA. The new model presented here, was forced by climatic data from three GCMs, and included a parameterized solar radiation term in order to compute glacier ablation.

Our results are based on a multi-model mean annual ELA, expressed as the average ELA from three CMIP5 models. We obtained a regional long-term mean annual ELA equal to 3775 m a.s.l. which is lower than the regional value (4083 m a.s.l.) under present climate conditions estimated by Carrasco et al. (2005, 2008). The three GCMs correctly reproduce the latitudinal gradient, showing an increase of the ELA from south to north across the MA.

The temporal variability of our modeled regional annual ELA shows some notable periods with below/above average ELA, lasting between two to almost five decades. In a regional context, periods of anomalously high ELA values in the MA (associated with negative glacier mass balance) are identified during 1550-1575 CE. By contrast, periods with consistent values below the long-term ELA mean are found between 1640 and 1670 and again between 1800 and 1848 CE.

Spectral properties of our modeled mean annual ELA reveal significant periodicities of 2 - 7 years, and significant signals associated with decadal and multi-decadal variability. Interannual SST variability in the El Niño 3.4 region showed strong and significant correlations with the mean annual ELA in each model. Furthermore, significant relationships were identified on multidecadal time scales, revealing an association between the mean annual ELA and multidecadal SST variability in the Pacific. This Pacific SST forcing was likely a major driver modulating of annual- to decadal-scale ELA behavior during 1500 - 1848 CE in MA region of South America.

Future studies should evaluate possible relationships between decadal to multi-decadal SST variability (e.g., Inter decadal Pacific Oscillation IPO; Pacific Decadal Oscillation PDO), and mass balance and ELA behavior in others regions of South America that remain unexplored in terms of glacier-climate dynamics.





## Appendix A: Solar radiation and albedo parametrization

### A1  Incoming solar radiation

We calculated incoming solar radiation $G$ following Annandale et al. (2002) and considering minimum and maximum daily
temperatures in the parameterization. We estimated the extraterrestrial radiation ($R_a$ in MJ $\cdot$ m$^{-2}$ $\cdot$ d$^{-1}$ units) following the
equations given by Duffie and Beckman (2013).

$$R_a = \left(\frac{24 \cdot 60}{\pi}\right)(G_{cs})(dr)\left[\omega_s \sin(\phi)\sin(\delta) + \cos(\delta)\cos(\phi)\sin(\omega_s)\right] \tag{A1}$$

Where $\pi = 3.1415...$, $G_{cs}$ is the solar constant (1367 W $\cdot$ m$^{-2}$ or 0,082 MJ $\cdot$ m$^{-2}$). The $dr$ term represents the correction for
the eccentricity of Earth's orbit around the sun on day $d$ of the year, considering $d$ in the Julian calendar. The $dr$ is calculated
as:

$$dr = 1 + 0.033 \cdot \cos\left(\frac{2\pi}{365} \cdot d\right) \tag{A2}$$

$\omega_s$ = sunrise hour angle (radians):

$$\omega_s = \arccos(-\tan(\phi)\tan(\delta)) \tag{A3}$$

$\delta$ = declination of the sun above the celestial equator in radians on day $d$ of the year ($d$ in the Julian calendar).

$$\delta = \left(\frac{23.4 \cdot \pi}{180}\right) \cdot \sin\left(2\pi\left(\frac{284 + d}{365}\right)\right) \tag{A4}$$

The $\phi$ angle (in radians units) represent the latitudinal location. We used a mean latitude of 33.50°S as reference latitude.
The computation of $\phi$ was calculated as:

$$\phi = \text{lat} \cdot \left(\frac{\pi}{180}\right) \tag{A5}$$

We calculated $G$ following the relationship described in Annandale et al. (2002), where $G$ is estimated as the product of
$R_a$ for $T_t$ ($G = R_a \cdot T_t$). The $T_t$ term represents the atmospheric transmissivity estimated based on Hargreaves and Samani
(1982), where $T_t$ for a given day is proportional to the square root of the difference between maximum temperature ($T_{max}$)
and minimum temperature ($T_{max}$). Furthermore, we considered a modified $T_t$ by the elevation ($Z$; in meters) provided in
Annandale et al. (2002), and expressed as:

$$T_t = K_{rs} \cdot (1 + Z \cdot 2.7 \cdot 10^{-5})(T_{max} - T_{min})^{0.5} \tag{A6}$$





The $K_{rs}$ is an adjustment coefficient for interior or coastal regions. In our case, we used the 0.16 for interior locations where a land mass dominates. We considered as Z the mean elevation based on the topography obtained from the used grids by each respective GCM.

## A2   Albedo parametrization

We calculated albedo using daily precipitation, and following the relationships describe by Oerlemans and Knap (1998). We
used snowfall events related to more precipitation. We considered that albedo of the glacier surface in a "d" day depends on the age of the snow at the surface. Their relationship is described as:

$$\alpha_s^t = \alpha_{fi} + (\alpha_{fr} - \alpha_{fi}) \cdot e^{-\Delta t/t^*} \tag{A7}$$

$$\alpha^t = \alpha_s^t + (\alpha_{hielo} - \alpha_s^t) \cdot e^{-d/d^*} \tag{A8}$$

The $\alpha^t$ corresponds to the global albedo global at the surface in a specific $t$ day. $\alpha_s^t$ corresponds to the snow albedo at the
surface in a $t$ day. The $\alpha_{fr}$ and $\alpha_{fi}$ parameters are related to fresh snow albedo and firn or old snow albedo, respectively. $\alpha_{hielo}$ represent a specific glacier ice albedo, while $t^*$ corresponds the temporal scales that represent the transition of fresh snow albedo to firm. The $\Delta t$ term is referred to days from the snowfall event. The $d$ and $d^*$ parameters correspond to the snow depth (in meters), and scale coefficient of snow depth, respectively. When the depth is $d^*$, the snow contribution is $1/e$ to the total albedo. The list of values used to estimate albedo is summarized in Table A1.

**Table A1. Summary of parameters used to calculated albedo from GCMs precipitation data. References values are obtained from Paterson (1994)**

| abbreviation | Parameter | Value |
|---|---|---|
| $\alpha^t$ | Global daily albedo at surface in a $t$ day | |
| $\alpha_s^t$ | Snow albedo at surface in a $t$ day | |
| $\alpha_{fr}$ | Fresh snow albedo | 0.85 |
| $\alpha_{fi}$ | Firn or snowpack albedo | 0.60 |
| $\alpha_{hielo}$ | Ice albedo | 0.30 |
| $t^*$ | Time between transition of snow albedo to old snow albedo | 3 days |
| $\Delta t$ | Days from the last snowfall event | |
| $d$ | Snow depth (in meters) | |
| $d^*$ | Scale coefficient of snow depth | 0.4 |



*Competing interests.* The authors declare that they have no conflict of interest.

*Acknowledgements.*

Álvaro González-Reyes wishes to thank Comisión Nacional de Investigación Científica y Tecnológica–Programa de Capital Humano Avanzado (CONICYT-PCHA)/Doctorado Nacional/ 2016-21160642 for the doctoral scholarship. C. Bravo acknowledges support from the CONICYT Programa Becas de Doctorado en el Extranjero, Beca Chile, for the doctoral scholarship. We acknowledge support from Fondo de

Financiamiento de Centros de Investigación en Áreas Prioritarias (FONDAP) 15110009 [Center for Climate and Resilience Research [(CR)2] and Advanced Mining Technology Center AMTC of Universidad de Chile.



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





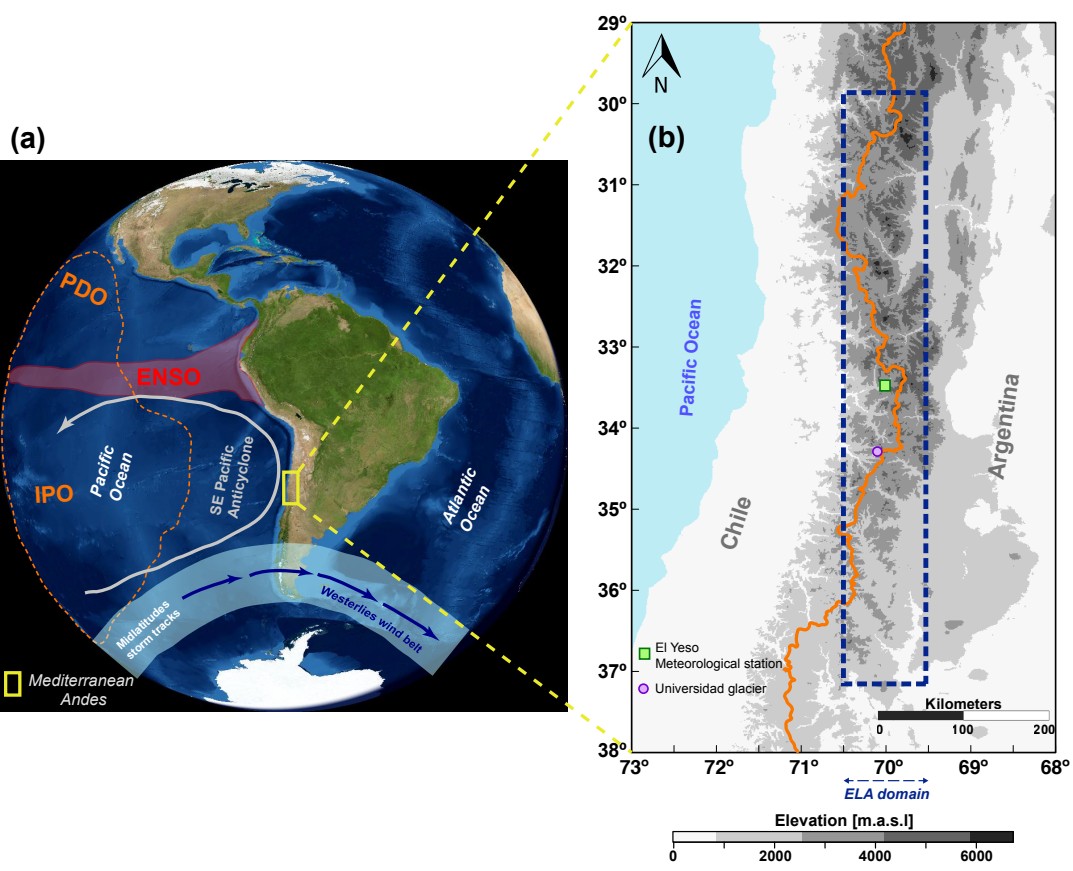

**Figure 1.** Schematic representation of the study area. a) The main large-scale climate forcings that interact over the Mediterranean Andes region, and b) the topography of the MA region based on Shuttle Radar Topography Mission – Digital Elevation Model (SRTM-DEM) data with 90 m resolution. The approximate domain where the ELA has been estimated is shown with dashed blue lines. The square and dot symbols represent the El Yeso meteorological station and Universidad glacier, respectively.





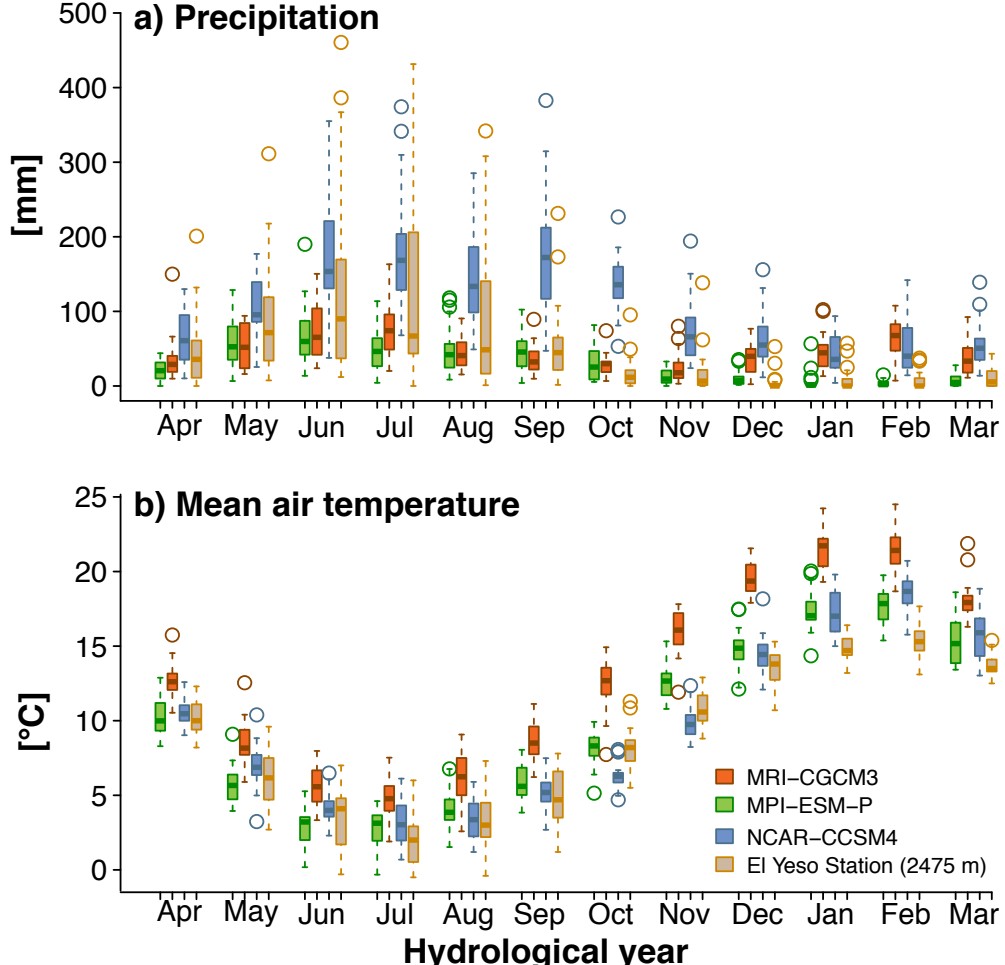

**Figure 2.** Comparison (boxplots) of a) monthly mean precipitation (in mm) and b) monthly mean air temperature at 2 m (in C) between observations at station YESO and corresponding grid cell from three GCMs. Model data was interpolated to equivalent station elevation of 2475 m. Data is based on average over period 1979-2010.



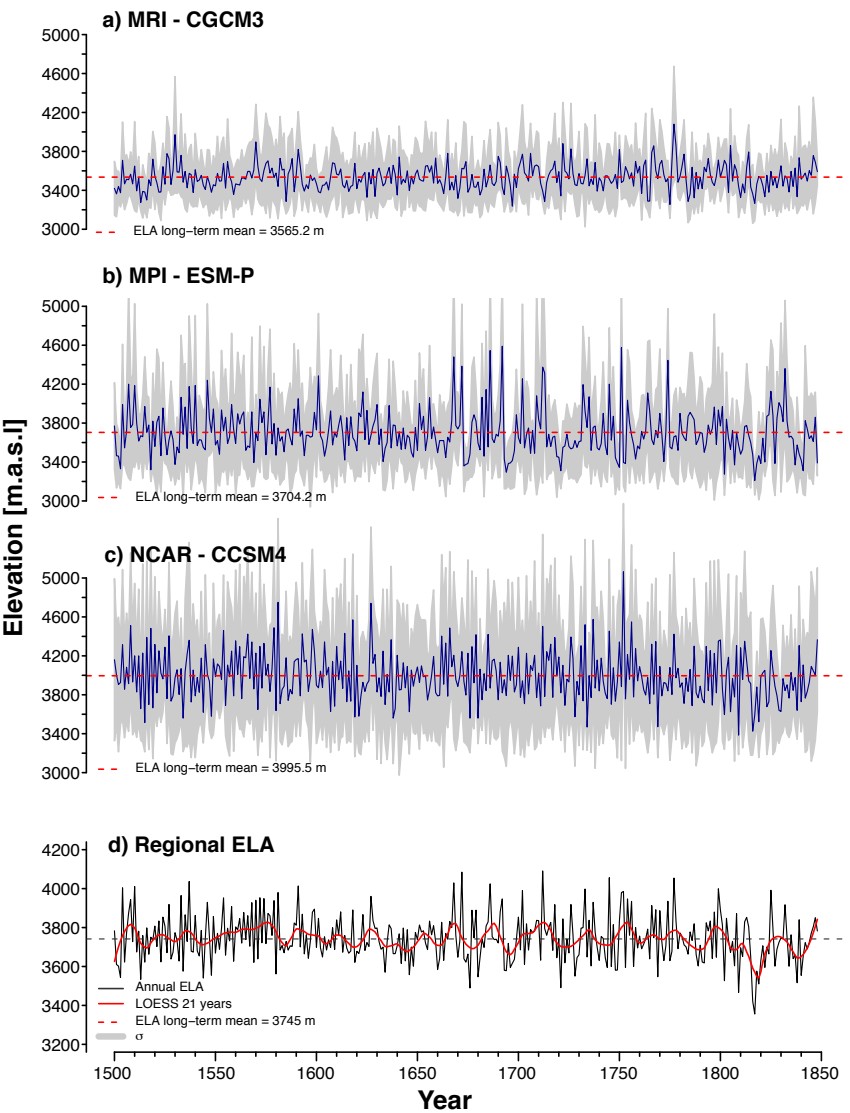

**Figure 3.** Annual ELA time series spanning the period 1500 - 1848 CE expressed as meters above sea level (m.a.s.l) based on a) three CMIP5 historical simulations. The annual year corresponds to the hydrologic year starting in April and ending in March of the following year. The long-term mean of the regional ELA, calculated for the entire period, is shown as a red dashed line in d). The grey contours represent $\pm standard deviation calculated for the period 1500-1848 CE$.

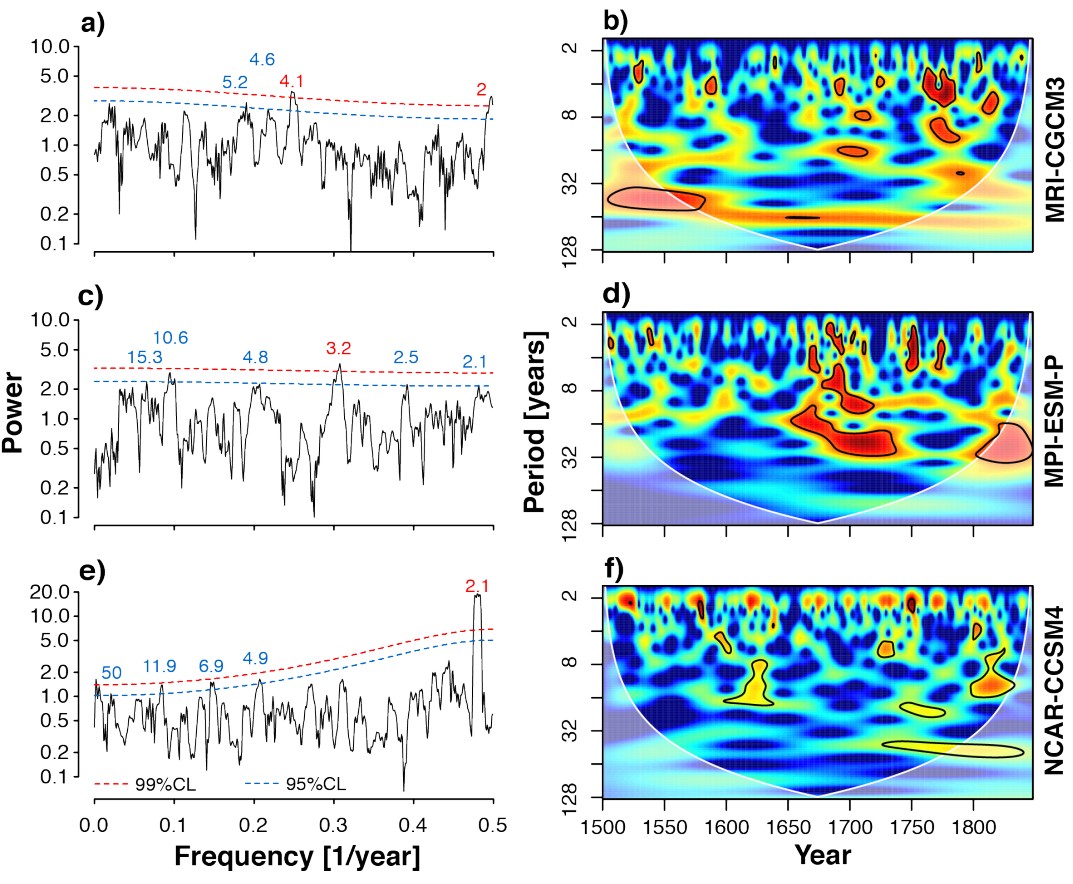

**Figure 4.** MTM Spectral density of mean annual ELAs from the models: (a) MRI-CCSM3, (c) MPI-ESM-P and (e) NCAR-CCSM4 GCMs. Numbers in blue and red indicate cycles (in years) statistically significant at P < 0.05 and 0.01, respectively. Analogue to (a), (c) and (e) panels, the (b), (d) and (e) panels represent the continuous wavelet transform (CWT) using a Morlet adjustment to detect the main cycles contain of each regional annual ELA. The thick black contours within the cone of influence designate the confidence level using a red noise model at P < 0.05. The areas beyond is shown as a lighter shade.







**Figure 5.** Cross wavelet and wavelet coherence between each regional annual ELA and PC1 of April to March SSTA over the Niño 3.4 region, obtained via EOFs from the respective GCM. The (a), (c) and (e) panels represent the cross wavelet, while (b), (d) and (f) show the wavelet coherence analysis. The arrows within signals with black contours of each panel indicate the two time series vary in-phase or anti-phased. The thick black contours indicate the significance level at P < 0.05 obtained based on a red noise model. Areas outside the cone of influence are shown in a lighter shade.



**Figure 6.** Maps of Pearson's correlation coefficients calculated over 1500-1848 CE between mean SST anomalies (averaged from April to March) and the respective mean annual ELA time series. (a,c,e) correlation of interannual variability, (b,d,f) correlation on multi-decadal time scales using a 21-year running LOESS filter. The SST datasets correspond to the models MRI-CGCM3, MPI-ESM-P and NCAR-CCSM4. The r Pearson's correlation values of |0.11| present statistical significance at P < 0.05 using year to year data, while that values of |0.43| are statistically significant at P < 0.05 using a 21-year running LOESS.

Appendix figures




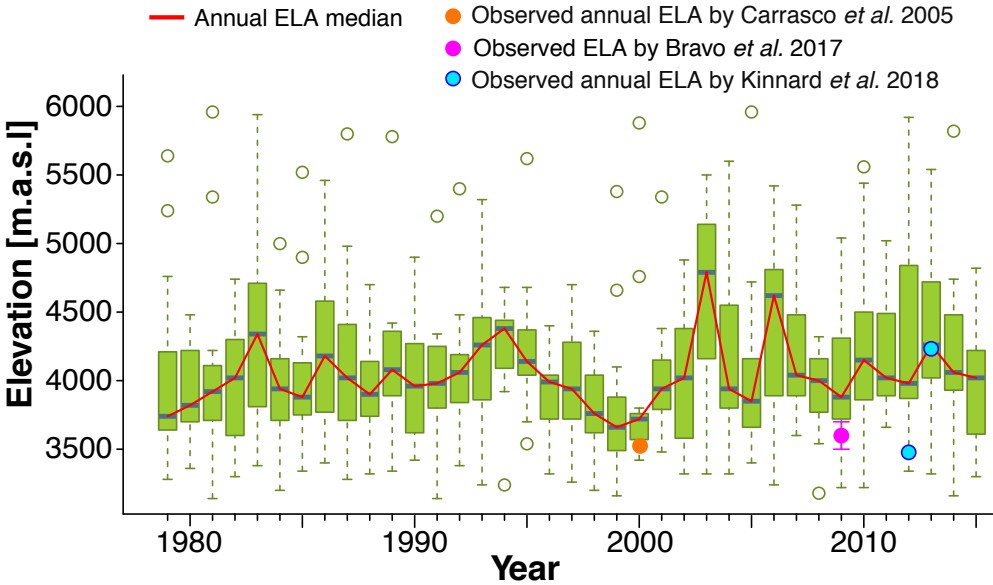

**Figure A1.** Equilibrium line altitude (ELA) of Universidad Glacier (34°40'S, 70°20'W) during 1979 - 2016 period. The glaciological model has been forced by a grided product of daily precipitation, and mean, minimun and maximun daily temperatures obtained from Chilean meteorological stations. The spatial grid resolution by grid is 5-km, and the data has been compiled by Climate and Resilience center CR2. The data can be downloaded freely from: http://www.cr2.cl/datos-productos-grillados/?cp_cr2met=2

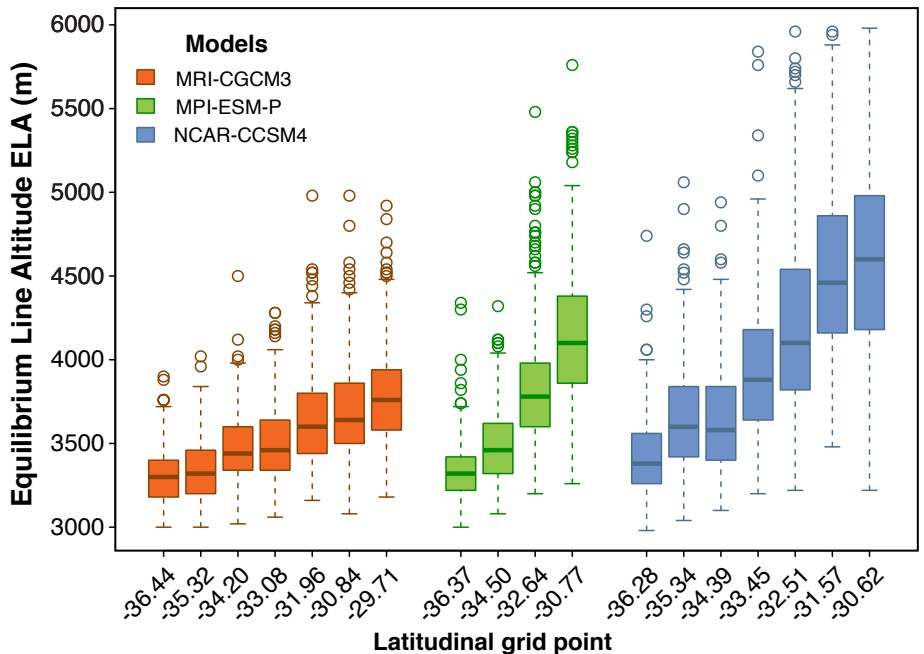

**Figure A2.** Summary of modeled equilibrium line altitude ELA from each grid by GCM. The x-axis represent the mean latitude of each grid.





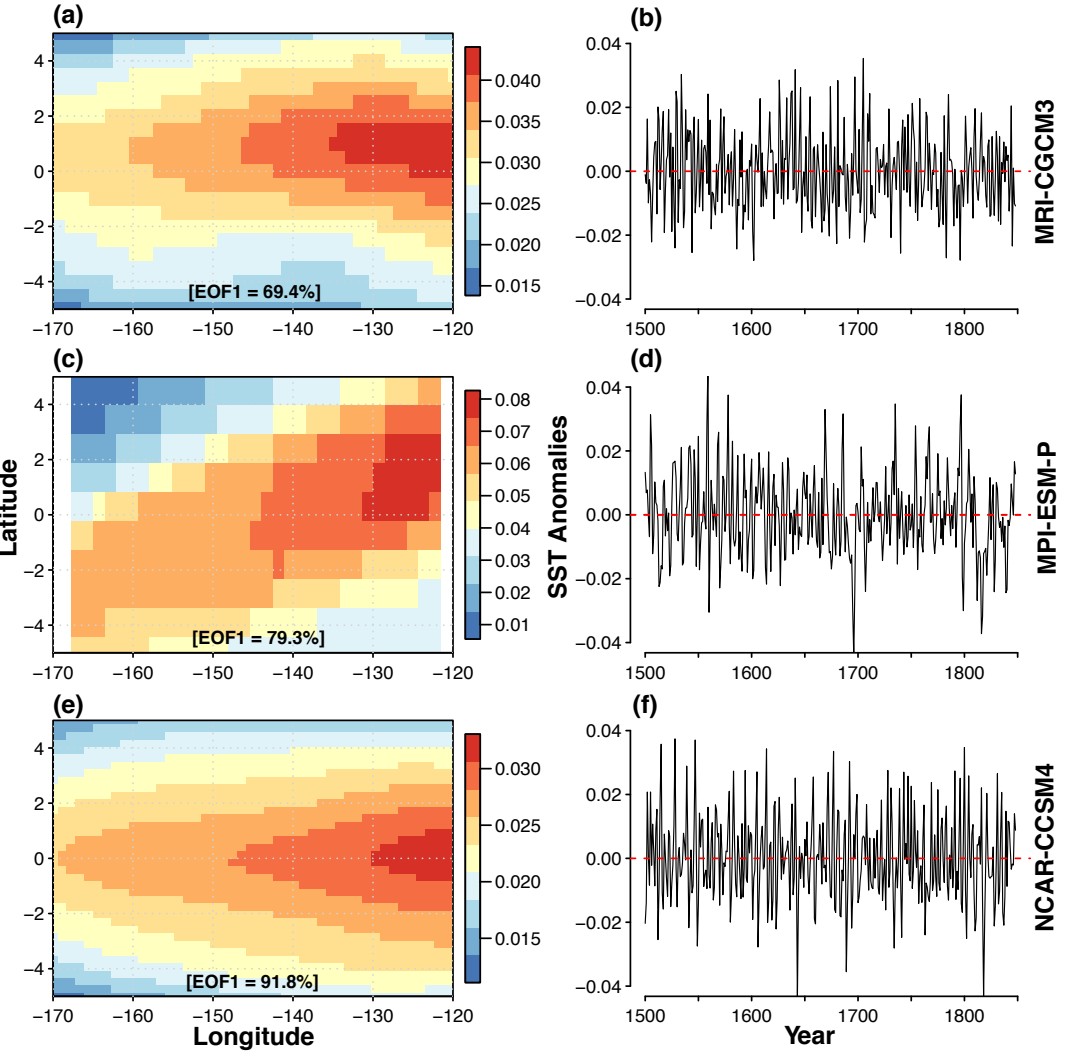

**Figure A3.** Summary of the leading Empirical Orthogonal Functions (EOF) calculated over the SSTa over Niño 3.4 region from April to March months, and using data from three GCMs: MRI-CCSM3, MPI-ESM-P and NCAR-CCSM4 models. The spatial structure of the leading EOF is showed in a, c and e panels, and the % values represent the explained variance of the leading EOF respect to the total. The temporal amplitude of the EOF is showed in b, d and f panels. Each row shows the spatial and temporal structure of each EOF by GCM. The GCM names are indicate in the right y-axis.



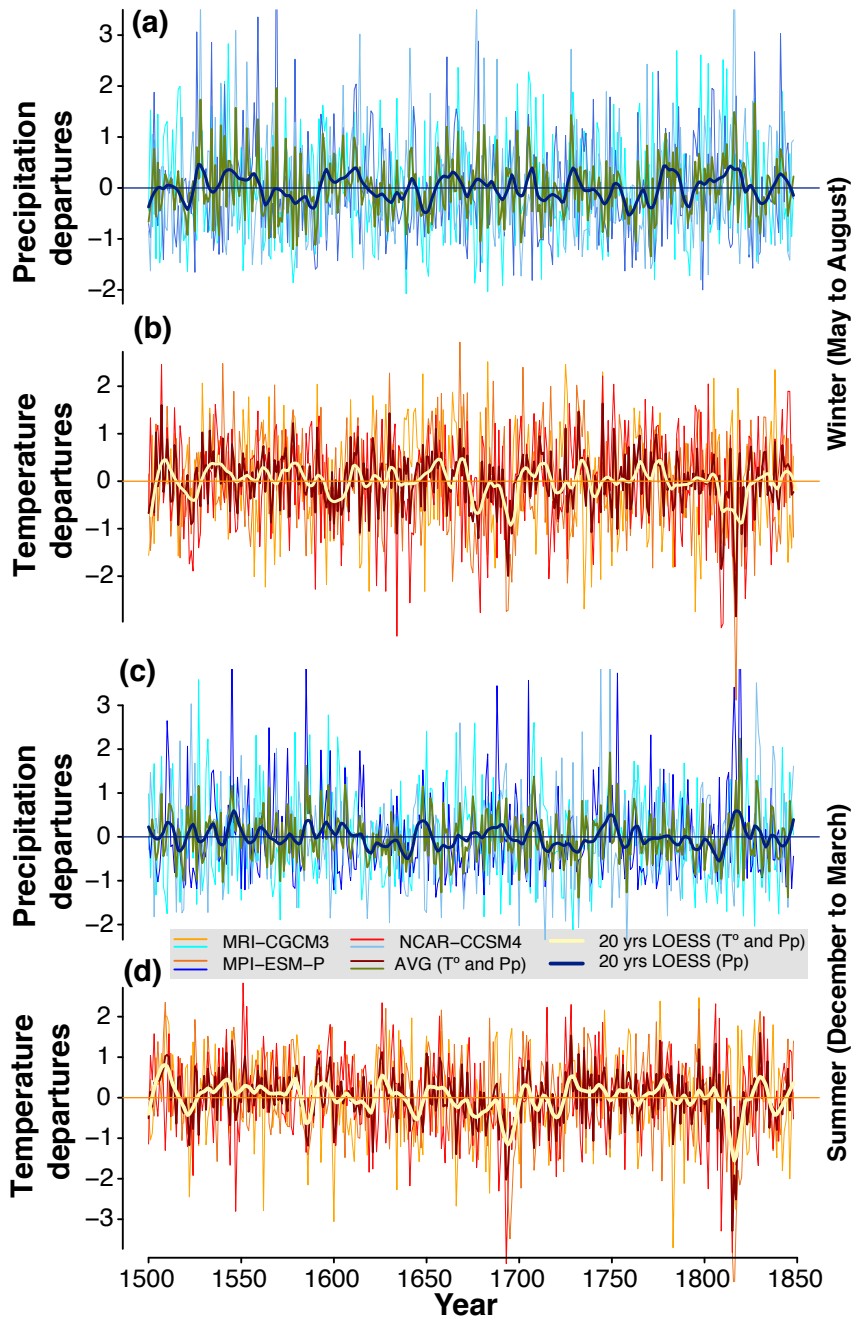

**Figure A4.** a) Regional precipitation during May to August months calculated as an average from GCMs expressed in standardized anomalies. The same as (a), but for mean air temperature expressed in standardized anomalies. The (c) and (d) panels represent the temporal variability of December to March months of total precipitation and mean air temperature, respectively. Both variables are expressed in standardized anomalies. In each panel a 20 - year LOESS filter has been used to highlight multi-decadal variability contain in the total regional precipitation and mean air temperature time series during year seasons.

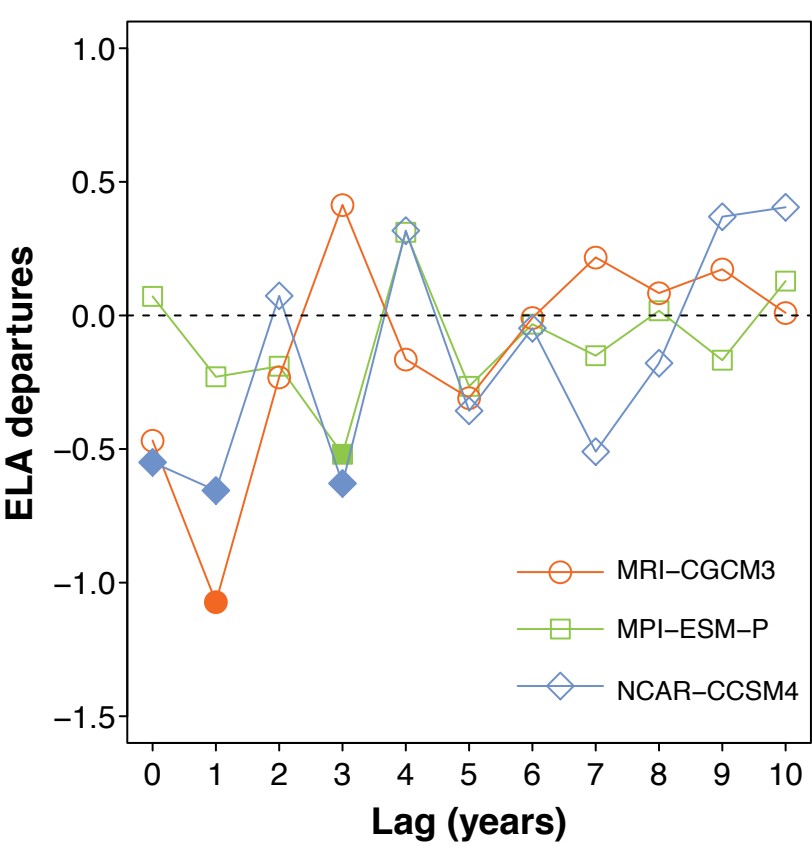

**Figure A5.** Superposed Epoch Analysis (SEA) between mean annual ELAs from GCMs and historical volcanic eruptions ocurred during the years: 1585, 1601, 1641, 1698, 1783 and 1816. The symbols with fill colors present statistical significant at $P < 0.10$

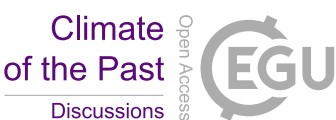

**Table 1.** Summary of each global climate model (GCM) used in this study.

| Model | Institute (Country) | Atmos Res (lat x long x levels) | Ocean Res (lat x long x levels) | References |
|---|---|---|---|---|
| MRI-CGCM3 | MRI (Japan) | 320x160 x L48 | 364x368xL51 | Yukimoto et al. (2011) |
| MPI-ESM-P | MPI (Germany) | 196x98 x L47 | 256x220xL40 | Raddatz et al. (2007) |
| NCAR-CCSM4 | NCAR (USA) | 288x192 x L26 | 320x384xL60 | Gent et al. (2011) |