# Peer review of "Glacier equilibrium line altitude variations during the "Little Ice Age" in the Mediterranean Andes $(30^{\circ} - 37^{\circ}S)$"

_Climate of the Past, 2019_

## Referee Comment (RC1) · Anonymous Referee #1 · 20 May 2019

General comments (overall quality)

This manuscript is an innovative research on the ELA evolution in the Mediterranean Andes (MA) during the 1500-1850 period. Past ELA modelling, including Little Ice Age (LIA) readvances, are scarce in the MA so this could be a significant contribution. The ELA reconstruction is based on state of the art, enhanced-temperature index model, adjusted with available ELA data and run with the input of three GCM. The model setup and data input are adequately described.

Nevertheless, I think the climate-ELA relationship for this region is excessively simplified. The ELA is a glaciological parameter, not a climatological one. Although it can,

and has been approximated with climatic analysis, it should always relate to glaciological observations (i.e.: glaciological mass balance, end-of–summer snowline, geomorphological evidence...). Accumulation and ablation are driven by climate but also strongly controlled by a large set of factors (glacier size, shape, dynamics, geomorphometry, climatic regime). In particular, I consider that the modelling exercise of this manuscript is loosely adjusted to four ELA observations in the largest and thus non-representative glacier of the MA for calibration, and to one for validation. If this point is not addressed I do not see how the conclusions can be generalized to a vast geographic domain encompassing thousands of very different ice masses.

More generally, the ELA should be considered with some caution in this region where, since the pioneer studies, it was acknowledged that some of these glaciers act as reservoir glaciers so the ELA can be, depending on the years, entirely over or under the existing glaciers (Lliboutry, 1965). This large variability, mostly attributed to precipitation (Rabatel and others, 2011; Masiokas and others, 2016; Farías-Barahona and others, 2019) might explain why some of the longest glaciological mass balance series do not provide information of a key glaciological variable such as ELA (Escobar and others, 1995; Leiva and others, 2007).

Therefore, I consider this manuscript should undergo major revisions prior to publication. For this I propose two alternatives. Either a larger set of calibration and validation data is produced/compiled, which is more according to the vast geographic domain proposed by the authors, but it might be out of the scope of this research piece. Or the modelling effort and its conclusions should be narrowed to an area or a set of glaciers where an acceptable calibration/validation data set exists.

Specific comments

Regarding calibration, only four ELA observations are provided for one glacier (Universidad) and they do not seem to fit the modelled data very convincingly (Fig A1), with 3 out 4 observations falling outside of the modelled error bars and errors being

not systematic. These differences range from almost nothing to hundreds of meters, which is even more disturbing, considering that Universidad is the largest glacier in the region and the several thousand other glaciers have a much lesser size and elevation range (Barcaza and others, 2017; Zalazar and others, 2017). There are some additional glaciological mass balance observations series in the MA (i.e: WGMS, 2017) and, although little ELA information is provided, more could be done to use available data. For instance, the longstanding Echaurren Norte data could be used to verify if modeled ELA follows measured mass balance since 1978. Additionally, end of summer snowline elevation for some glaciers could be used to validate modelled ELA of the past decades. This would provide a more robust estimation of the ability of the model to capture the current ELA behaviour over the entire target region (MA). Only once this is first step is achieved, is it worthy to try and reconstruct former regional ELA.

Also, while enhanced temperature-index might be suitable for large temperate glaciers in the Central Andes (31-35°S), where sublimation accounts only for 1-6 % of ablation (Pellicciotti and others, 2008; Kinnard and others, 2018), it does not appear suitable for the smaller cold glaciers both in the Central and Desert Andes (19-31°S) where sublimation can account for 80 % of ablation so energy balance approach should be used instead (Pellicciotti and others, 2008; MacDonell and others, 2013; Kinnard and others, 2018). Further, mass balance observations in the Desert Andes show little dependency of ablation and accumulation to elevation (Rabatel and others, 2011 in particular fig 6) so the use of vertical gradients to account for this processes is a major simplification which could lead to important uncertainties, especially in smaller glaciers. It also raises questions as to how to estimate ELA when it is not directly observed on glaciers.

Regarding the validation, one possible approach could be to compare past modelled ELA with geomorphologically-derived ELA in some sites. Medium altitude of present glacier (Leonard and Fountain, 2003) and highest elevation of lateral moraines of past glaciers could also be used as climatic ELA proxy for LIA glacier extent. Dated

moraines were used to constrain climate modelling for the LGM (Zech and others, 2011). Of course the main limitation, as mentioned by the authors, is the lack of late neoglacial chronologies in the region (i.e. dated moraines) for the 1500-1850 period. But, without them, it is hard to assess the performance of the ELA reconstruction and thus the validity of the ELA behavior analysis during the LIA.

Technical corrections

Line 475. Include the chapter tittle in the Kinnard et al 2018 reference. This is a key reference and only the book tittle is given.

Figure 2. Please indicate what the boxes, dashed lines and circles indicate (percentile confidence intervals?). Include this information in the other box plots of the manuscript (Figs A1 and A2).

Figure 3. According to recent inventories, modern medium elevation (climatic ELA proxy) for all debris free ice masses of MA is close to 4300 m asl, ranging from over 2400 to over 6500 m asl. This figure does not accurately represent the present value or the large variability of probable modern ELA.

Figure A1. This is important calibration data and should be moved to main section.

References

Barcaza G, Nussbaumer S, Tapia G, Valdés J, García J-L, Videla Y, Albornoz A and Arias V (2017) Glacier inventory and recent glacier variations in the Andes of Chile, South America. Ann. Glaciol.

Escobar F, Casassa G and Pozo V (1995) Variaciones de un glaciar de montaña en los Andes de Chile Central en las últimas dos décadas. Bull. Inst. Fr. DÉtudes Andin. 24, 683–695

Farías-Barahona D, Vivero S, Casassa G, Schaefer M, Burger F, Seehaus T, Iribarren-Anacona P, Escobar F and Braun M (2019) Geodetic Mass Balances and Area

[Figure]

Changes of Echaurren Norte Glacier (Central Andes, Chile) between 1955 and 2015. Remote Sens. 11(3), 260 (doi:10.3390/rs11030260)

Kinnard C, MacDonell S, Petlicki M, Mendoza Martinez C, Abermann J and Urrutia, Roberto (2018) Mass balance and meteorological conditions at Universidad glacier, Central Chile. Andean hydrolgy. CRC Press, 102–126

Leiva JC, Cabrera G and Lenzano L (2007) 20 years of mass balances on the Piloto glacier, Las Cuevas river basin, Mendoza, Argentina. Glob. Planet. Change 59, 10–16

Leonard KC and Fountain AG (2003) Map-based methods for estimating glacier equilibrium-line altitudes. J. Glaciol. 49, 329–336

Lliboutry L (1965) Traité de glaciologie. Tome II. Glaciers, variations du climat, sols gelés. Masson & Cie, Editeurs, Paris

MacDonell S, Kinnard C, Mölg T, Nicholson L and Abermann J (2013) Meteorological drivers of ablation processes on a cold glacier in the semi-arid Andes of Chile. The Cryosphere 7(5), 1513–1526 (doi:10.5194/tc-7-1513-2013)

Masiokas MH, Christie DA, Le Quesne C, Pitte P, Ruiz L, Villalba R, Luckman BH, Berthier E, Nussbaumer SU, González-Reyes Á, McPhee J and Barcaza G (2016) Reconstructing the annual mass balance of the Echaurren Norte glacier (Central Andes, 33.5° S) using local and regional hydroclimatic data. The Cryosphere 10(2), 927–940 (doi:10.5194/tc-10-927-2016)

Pellicciotti F, Helbing J, Rivera A, Favier V, Corripio J, Araos J, Sicart J-E and Carenzo M (2008) A study of the energy balance and melt regime on Juncal Norte Glacier, semi-arid Andes of central Chile, using melt models of different complexity. Hydrol. Process. 22, 3980–3997 (doi:10.1002/hyp.7085)

Rabatel A, Castelbrunet H, Favier V, Nicholson L and Kinnard C (2011) Glacier changes in the Pascua-Lama region, Chilean Andes (29oS): recent mass-balance and 50 year surface-area variation. The Cryosphere 5, 1029–1041

[Figure]

WGMS (2017) Global glacier change Bulletin. Bulletin No. 2 (2014-2015). WGMS, Zurich, Switzerland

Zalazar L, Ferri Hidalgo L, Castro M, Gargantini H, Gimenéz M, Pitte P, Ruiz L, Masiokas M and Villalba R (2017) Glaciares de Argentina: resultados preliminares del Inventario Nacional de Glaciares. Rev. Glaciares Ecosistemas Mont. 2, 13–22

Zech R, Zech J, Kull C, Kubik PW and Veit H (2011) Early last glacial maximum in the southern Central Andes reveals northward shift of the westerlies at ∼39 ka. Clim. Past 7, 41–46 (doi:10.5194/cp-7-41-2011)

---

## Referee Comment (RC2) · Anonymous Referee #2 · 10 Jul 2019

**1   GENERAL COMMENTS**

The manuscript tackles the very interesting question of the influence (if any) of the LIA experienced in the Northern Hemisphere between 1500-1850 CE on the Mediterranean Andes. Little research has been published about this subject, making this line of research of great interest. The authors use a novel and promising method based on the derivation of glaciological parameters from GCM reconstructions. This approach could provide relevant insight into the climatic anomalies during the Little Ice Age in the Mediterranean Andes, and the effect of such anomalies on glacier mass balance.

Although the title suggests that the authors will set the focus on the climatic variations during the LIA, the abstract and introduction explain that serious doubts exist about the very existence of an LIA climatic anomaly on the MA. Therefore, the main question becomes whether there is or not an LIA climatic anomaly on the MA. However, the rest of the paper focuses on the climatic analysis of the LIA itself, not allowing the comparison of the LIA with the centuries immediately before and after it. It would make sense to perform the ELA calculation over the whole millennia covered by the GCMs used, or at least starting a century or two before the LIA and extending it to the present. In such a way, it would be possible to establish whether the ELA during the LIA was anomalously low or not.

Using such a novel method, the authors should put more effort into the validation of the outputs. The comparison of the ELA time series derived from CR2 data with Universidad Glacier studies is an interesting exercise, but there are only four data points, and the fit is poor and with non-systematic differences. Arguably, the aim of the method is not to reproduce the ELA of a specific glacier, but to use the ELA as a proxy for regional glacier Mass Balance. Therefore, the misfit with the Universidad Glacier ELA would not be a problem, but it would also be of little help to assess the performance of the method (especially because the authors made this comparison using CR2 data instead of the past1000 runs of the CMIP5). In general, a more significant effort needs to be made to validate the methodology. For this, it would be advisable to extend the computation of the ELA to the present using the past1000 runs and compare it with a compilation of the mass balance data available in the MA. It would be reasonable to also apply the method over the Alps using the same CMIP5 runs and using the wealth of validation data available in that area. This Northern Hemisphere test would also allow seeing how well this method can identify the LIA anomaly in an area where its existence is well established.

Also, the authors should acknowledge the significant errors that might arise from the simplified model used to compute the ELA, but emphasizing that they aim to capture

the relative variations of the ELA in time, not to compute exact past absolute values.

In some instances, it appears as the authors are picking features of one or another model to support an assortment of ideas. However, they avoid an honest discussion of the substantial differences between GCMs, and the implications of such differences in terms of reliability.

Finally, much clarity could be gain with less liberal use of the concept of "Equilibrium Line Altitude". Often the manuscript refers to "daily" and "seasonal" ELA, while this concept is valid, by definition, only at an annual timescale. The method to compute the ELA is explained in a very superficial way. However, the reader can guess that they determine the ELA as the elevation at which mass balance is zero over any given time interval. However, when the authors compute a daily "ELA", it is unclear which value should it take in days where mass balance is above/below zero at all elevations (for example days with positive ablation at all elevations and no accumulation). The manuscript will be easier to understand if the term ELA is reserved only for annual timescales. The references to "daily" and "seasonal" ELAs does not seem to be necessary at all and can be avoided without any loss of clarity. Moreover, even using the concept of ELA at an annual timescale, the authors should explain the significant differences between the ELA they compute, and the common glaciological meaning of it, which is associated to a specific glacier, and influenced by mass advection, wind redistribution of snow, local topography, local winds, etc.

Summarizing, this manuscript has the potential to be an excellent piece of research, providing a very interesting contribution. While it could be published with minor revisions, I strongly suggest doing some major ones to address concerns regarding validation and the time interval over which the ELA was calculated.

**2 SPECIFIC COMMENTS**

Line 12

"during the period 1500 – 1848 CE" At this point, I would just say "during the LIA", because the fact that you do not use the period just used to define the LIA (1500 – 1850 CE) is distracting, and the reason for this becomes clear much later, and it is not relevant in the abstract.

Line 17

The acronym for empirical orthogonal functions should be spelled out here as it is the first mention of it. Also, for "EOF1" one would understand that you are referring to the first EOF. Therefore, "first EOF1" would be redundant.

Line 43

Include a short sentence saying what the large-scale estimate by Neukom et al. (2014) suggests. Does it show a temperature anomaly in the MA during the LIA period?

Line 95

"portion" suggests that you are doing a sub-diurnal analysis and that you compute the fraction of the total daily precipitation falling as snow. However, Eq. 2 suggests that you either consider all or none of the daily precipitation as snow.

Line 106

The authors should say here that G(t) is not a variable available directly as a GCM model output, and it is instead estimated using the parametrization by Annandale et al. (2002), that is based on the minimum and maximum daily temperatures, as well as the relative position to the Sun. Refer to Appendix A1 for more details.

Line 107

Although it is clear what you mean by snowpack here, the terminology is vague and not consistent throughout the manuscript. I would suggest referring to this consistently as firn.

Line 107

You should add here that the actual albedo is calculated following Oerlemans and Knap (1998), a method that takes into account snow age and snow depth, and refer to Appendix A2 for details.

Line 110

Who obtained those factors? Pellicciotti? (be explicit). What do you mean by "based". Is that the average? How different are the values at Juncal Norte and San Francisco? Give a sense of the variability of those factors along the MA.

Line 112

How similar?

Line 115

Eq. 3 say $T_{mean} > T_{crit}$, and here you say $T_{mean} > 0$ . If the $T_{crit}$ of Eq.2 and Eq.3 refer to different parameters, please use a different symbol. Or just write $T_{mean} > 0$ in Eq.3

Line 125

"GCMs based on past1000 experiment simulations (runs r1i1p1) of the CMIP5 initiative". Explain what the "past1000" refers to, and give some detail of those CMIP5 runs. Otherwise, the wording in this sentence is confusing.

Line 134

You refer to "this period" before actually defining it. Then define it in the following sentence. Please rephrase.

Line 135

You said in line 131 that "specific grid point information [of the GCM I guess] was used to compute temperature and precipitation lapse rates". However, here you say that you use a standard and constant lapse rate for temperature. And in line 139 you say that you also use a constant lapse rate for precipitation. Therefore, the statement in this line is misleading.

Line 136

You have not said why you need maximum and minimum temperature; they do not show up on any equations. This ambiguity will be solved if you introduce the suggestion made for line 106.

Line 140

In general, there is no such thing as a "daily ELA" or "winter ELA" or "summer ELA", see general comments. Nevertheless, this daily and seasonal values seem irrelevant as they are not used anywhere in the results or discussion.

Line 149

Is it well established that the first EOF of the SST captures the ENSO signal? If so, give a reference at least.

Line 150

Be consistent. PC1 or EOF1. Use either PC or EOF nomenclature.

Line 176

Again, it has not been explained why you need min/max temperatures to compute the ELA. See comments on line 136 and 106.

Line 177

It says "period 1979 – 2015" but Figure A1 says 1979-2016.

Line 186

How is the range of modeled ELA calculated? How do you estimate the uncertainty?

Line 192

This section is confusing, especially at this sentence. It should be more explicit that the authors are testing the ELA calculation method using a completely different dataset than the one used during the LIA period. Then, it is unclear the relevance of this comparison between the Carrasco ELA and the authors mean LIA ELA.

Line 194

General comment to section 3.2: Is this the most appropriate way to compare/validate? Arguably, it would be more interesting to see how these models reproduce the inter-annual and decadal variability of YESO station.

Line 196

"quite well" is too succinct and not substantiated. NCAR seems, and MRI seems to overestimate summer temperatures significantly. Also, Figure 2 excessively aggregates the data. If we are looking for anomalies in a time series, it would be better to see the time series of El Yeso and the GCMs between 1979 and 2015, not just monthly means.

Line 207

General comment to section 3.3: The discussion of the difference between model's mean ELAs does not contribute much. The periods of high/low ELA are interesting, and it would be helpful if figure 3 is modified to show these periods better. Maybe a smoothed version of the data can be presented in all panels (not just panel d), and vertical lines can be incorporated to highlight the periods you talk about here.

Line 223

These "significant and common periodicities" were not obvious to me in figure 4. It would be interesting to highlight those period intervals in figure 4. Only the two year periods seem to be common to all models and maybe something around five years. A log-scale in the X-axis of figure 4 might help the visualization.

Line 226

They look quite different to me.

Line 236

"EOF1 retained". Maybe "explained" is better terminology.

Line 236

"of the total variance" of what?

Line 237

The authors again talk about a sub-annual ELA, which is confusing. See general comments.

Line 240

What about the mismatch in timing of the periods with joint periodicities? The authors also talk about periods in-phase and anti-phase, which is quite confusing, and they do not address this later.

Line 261

Where was this comparison on the results? It does not seem to be there. Or are the authors talking about the 1979-2010 comparison of figure 2? The latter is what line 263 seem to suggest.

Line 271

This comparison would be much more useful if the authors present the calculation of

the present ELA using the past1000 runs.

Line 279

The authors can not know if there was a lower/higher ELA during the LIA compared with the second half of the past millennium because they only computed the ELA during the LIA. To know if there was a climate anomaly that could generate a glacier advance in the MA during the LIA, they need to compute the ELA over the whole past1000 data range, or at least in a range that extends beyond the LIA. With that information, they could see how the ELA during the LIA compares with the ELA before and after it, and with the prediction for the present (using past1000 runs data).

Line 292

Again, it would be nice to have those periods clearly highlighted in figure A4. Otherwise, it is difficult to see what the authors mentioned here.

Line 299

Shouldn't this be in the results?

Line 313

It seems advisable to replace "values" with "anomalies" or "departures". Otherwise, the authors would be suggesting that there is such a thing as "negative precipitation".

Line 316

Where did the authors observed that? There is no such data in the results.

Line 327

Statement a bit too strong for the data support. For MRI-CGCM3 there does not seem to be a significant relationship.

Line 329

Perhaps it would be better to say "might mask". If the authors say "seems to mask", they should explain better why it seems that way.

Line 331

Consistent, but one would not expect to see such a quick response to climate. So better than saying "low ELA values around 1840" they could say it in the same way they do in the following paragraph: "low ELA between 1800-1848". The very low ELA they found around 1820 might have more to do with that maximum advance than the low ELA around 1840. Therefore, to associate the advance right away with the whole 1800-1848 period makes more sense.

Line 334

However, the authors can not say if it was colder than the centuries before or after that interval (1500-1848 CE), which is also very relevant for the discussion and the comparison with the northern hemisphere. A paragraph like this is missing in the conclusions.

Line 341

Again referring to a "daily ELA", see general comments.

Line 360

It would be very illustrative and helpful to have a figure with the mean time series of SST over the El Niño 3.4 region (and some visualization of the EOF1 of it) alongside the ELA derived from each model.

Line 385

"$R_a$ for $T_t$". Perhaps "times" is a better term than "for" here.

Line 387

$T_{max}$ should be $T_{min}$

Line 391

Why not use the elevation in the range 100-6000 m used for MB calculation? Please justify this choice.

Line 398 (Eq. A8)

It would be better to use $\alpha_{ice}$ instead of $\alpha_{hielo}$

Line 399

What does "global" mean in this context? Is it just the actual surface albedo?

Figure 5

Explain better how to interpret the arrows. The explanation in the caption is binary (in phase or anti-phase). However, the arrows can be seen in all directions. Do they display the angular phase difference?

Figure 6

It would be nice to have a box showing the El Niño 3.4 region as well as the MA.

Figure A1

What the do the bar sizes, error bars and green circles mean?

**3  TECHNICAL CORRECTIONS**

Line 57: No brackets around SST

Line 66: First sentence of the line is redundant.

Line 111: I would recommend writing "$90 \times 10^{-4}$ (mm $h^{-1}$ C)" instead of "90 (mm $h^{-1}$ C) $\times 10^{-4}$"

Line 114. Change "Appendix section" by "Appendix A"

Line 122: studies CARRIED out by Kinnard...

Line 147: Hyphen on long-term seem to have extra spaces or be a long hyphen.

Line 250: "MRI-ESM-P", I think it should be "MPI-ESM-P"

Line 260: "in terms to" might better be "in terms of"

Line 399: Word "global" is repeated

Line 402: "firm" should be "firn"

Figure 3: "$\pm$standarddeviationcalculatedfortheperiod". Better put it in words.

Figure 4: The areas beyond ARE shown as a lighter shade.

Figure 5: I guess that SSTA should be just SST

Figure A4: variability containED in the total regional precipitation

---

## Author Comment (AC1) · 29 Oct 2019

We greatly appreciate your comments and suggestions in order to improve our manuscript. Many thanks. In order to check our modeled annual ELA for the 1979 – 2015 period, we contrast it with ELA information at annual resolution, obtained from Landsat images for five glaciers located across the Mediterranean Andes region (Juncal Norte, Olivares Gamma, Cipreses, Cortaderal and Universidad). Landsat images (MSS, TM, ETM+, OLI) have been widely used to obtain snowlines on glaciers (Rabatel et al., 2012, 2013; Wastlhuber et al., 2017; Rastner et al., 2019). The free access to images and the high acquisition frequency allows us to count with a long-time coverage of many glaciers worldwide. To contrast our modeled ELA with observations, we used ten years for comparison within the 1986 – 2014 period due to availability of images by this Andean region. In the results section, we comment that our ELA model presents similitudes with the ELA values from Landsat images (Figure 1). Our results show a good ELA representation for Juncal Norte and Olivares Gamma glacier (Figure 1a), which are located around of 33°00'S and 70°10'W within the MA region. Our modeled annual median ELA shows congruence with annual ELA values derived from Landsat images within 1979 – 2015. In this location of the MA region, our modeled ELA reproduces well the annual average ELA condition. In the case of satellite-derived ELAs from Universidad, Cortaderal, and Cipreses glaciers, our model shows an overestimation (Figure 1b). We think that differences between hydroclimatic patterns over the MA region (i.e., annual precipitation distribution and spatio-temporal variability) are essential to explain these differences in both locations of MA. A previous study carried out by González-Reyes et al. 2017 exposes spatio-temporal hydroclimatic differences in the MA region. We think that such previously documented temporal ELA differences are associated with interannual variability due to ENSO in the northern portion of MA (30° - 33°S), while from 34°S southward, the multidecadal hydroclimate variability, e.g. precipitation, is associated with the Interdecadal Pacific Oscillation (IPO). In the new version of our manuscript, we added a full description on the annual ELA estimation methodology using Landsat images. In addition, we added a new table (see table 2) that summarizes the specific values of annual ELAs calculated for each glacier, along with the associated total error.

Technical specifications

Line 475. Include the chapter tittle in the Kinnard et al 2018 reference. This is a key reference and only the book tittle is given.

We included the chapter title of Christoph Kinnard and coauthors 2018 in the manuscript. Thanks.

Figure 2. Please indicate what the boxes, dashed lines and circles indicate (percentile confidence intervals?). Include this information in the other box plots of the manuscript (Figs A1 and A2).

We added a full description of the dashed lines and circles. Thanks.

Figure 3. According to recent inventories, modern medium elevation (climatic ELA proxy) for all debris free ice masses of MA is close to 4300 m asl, ranging from over 2400 to over 6500 m asl. This figure does not accurately represent the present value or the large variability of probable modern ELA.

This Figure shows the ELA results for each GCM used as input for our mass balance model. As the Reviewer noted, the resolution of each GCM does not allow to identify local ELAs, which also depend on local factors. The range of values mentioned by the Reviewer is related to these local factors. Hence, it refers to a spatial variability within the MA during a very specific time (year of the inventory), while our Figure shows a time series. As we mentioned in Page 6, Line 145, we used the concept of "Regional ELA" with the aim to identify the generalized response of the glaciers within the MA.

Bibliography

González-Reyes, Á., McPhee, J., Christie, D. A., Le Quesne, C., Szejner, P., Masiokas, M. H., Villalba, R., Muñoz, A. A., and Crespo, S.: Spatiotemporal Variations in Hydroclimate across the Mediterranean Andes (30° –37°S) since the Early Twentieth Century, J. Hydrometeorol., 18, 1929–1942, https://doi.org/10.1175/JHM-D-16-0004.1, http://journals.ametsoc.org/doi/10.1175/JHM-D-16-0004.1, 2017.

Rabatel, A., Bermejo, A., Loarte, E., Soruco, A., Gomez, J., Leonardini, G., Vincent, C., and Sicart, J. E.: Can the snowline be used as an indicator of the equilibrium line and mass balance for glaciers in the outer tropics?, Journal of Glaciology, 58, 1027–1036, https://doi.org/10.3189/2012JoG12J027, 2012.

Rabatel, A., Francou, B., Soruco, A., Gomez, J., Cáceres, B., Ceballos, J. L., Basantes,

R., Vuille, M., Sicart, J.-E., Huggel, C., Scheel, M., Lejeune, Y., Arnaud, Y., Collet, M., Condom, T., Consoli, G., Favier, V., Jomelli, V., Galarraga, R., Ginot, P., Maisincho, L., Mendoza,

J., Ménégoz, M., Ramirez, E., Ribstein, P., Suarez, W., Villacis, M., and Wagnon, P.: Current state of glaciers in the tropical Andes: a multi-century perspective on glacier evolution and climate change, The Cryosphere, 7, 81–102, https://doi.org/10.5194/tc-7-81-2013,https://www.the-cryosphere.net/7/81/2013/, 2013.

Rastner, P., Prinz, R., Notarnicola, C., Nicholson, L., Sailer, R., Schwaizer, G., and Paul, F.: On the Automated Mapping of Snow Cover on Glaciers and Calculation of Snow Line Altitudes from Multi-Temporal Landsat Data, Remote Sensing, 11, https://doi.org/10.3390/rs11121410, https://www.mdpi.com/2072-4292/11/12/1410, 2019.

Wastlhuber, R., Hock, R., Kienholz, C., and Braun, M.: Glacier Changes in the Susitna Basin, Alaska, USA, (1951–2015) using GIS and Remote Sensing Methods, Remote Sensing, 9, https://doi.org/10.3390/rs9050478, https://www.mdpi.com/2072-4292/9/5/478, 2017.
* * *
[Figure]

**Fig. 1.**

---

## Author Comment (AC2) · 4 Nov 2019

We greatly appreciate your comments and suggestions in order to improve our manuscript. Many thanks.

In our new version, we contrast our modeled annual ELA with annual ELA values for five glaciers located in the Mediterranean Andes region (MA; Juncal Norte, Olivares Gamma, Cipreses, Cortaderal and Universidad). We used this approach based on the scarce ELA information in the MA region. Now, we compared values for ten years out of the 1986 – 2014 period. We add a full description on this procedure into the methodology section. A summary of inferred annual ELAs is presented in table 2 and

shown in the new figure A1 in Appendix section (Here figure 3). In the new version of the manuscript, the results show a good representation of ELA in the Juncal Norte and Olivares Gamma glaciers (Figure 3a), located around 33°00'S and 70°10'W within the MA region. Our modeled annual median ELA shows congruence with annual ELA values derived from Landsat images for 1979 – 2015. In this location of the MA region, our modeled ELA reproduces the average observed values. For the ELAs obtained at Universidad, Cortaderal and Cipreses glaciers, our model shows an overestimation (Figure 3b). We think that these differences could be associated with, on the one hand, the precipitation gradient used (0.02 mm * mˆ-1). On the other hand, Gonzalez-Reyes et al (2017) report a precipitation gradient northward of 33.5°S and southward of 34°S. In addition, this gradient is clearly detected from the annual mean precipitation for 1979 - 2015, using the CR2met gridded precipitation dataset with 5km of horizontal resolution (see figure below). The same precipitation dataset has been used to run our mass balance model and to estimate our annual ELA in the MA region. In addition, in our new version of the manuscript, we highlight that the temporal approach to our ELA analysis is on the annual scale.

Given that the main focus of our research is the 1500 – 1850 CE period, we do not extend the analysis back in time. However, we appreciate your suggestion on the possibility to extend the analysis in order to include another relevant paleoclimatic period, such as Medieval Climate Anomaly to present days, as well as to include a comparison between the Alps and the Andes. In fact, in order to understand past ELA variability on the last millennia in both hemispheres, we plan to prepare a new manuscript in the future.

González-Reyes, Á., McPhee, J., Christie, D. A., Le Quesne, C., Szejner, P., Masiokas, M. H., Villalba, R., Muñoz, A. A., and Crespo, S.: Spatiotemporal Variations in Hydroclimate across the Mediterranean Andes (30° –37°S) since the Early Twentieth Century, J. Hydrometeorol., 18, 1929–1942, https://doi.org/10.1175/JHM-D-16-0004.1, http://journals. ametsoc.org/doi/10.1175/JHM-D-16-0004.1, 2017

Specific comments

Line 12 "during the period 1500 – 1848 CE" At this point, I would just say "during the LIA", because the fact that you do not use the period just used to define the LIA (1500 – 1850 CE) is distracting, and the reason for this becomes clear much later, and it is not relevant in the abstract.

We included "during the LIA". Thanks

Line 17 The acronym for empirical orthogonal functions should be spelled out here as it is the first mention of it. Also, for "EOF1" one would understand that you are referring to the first EOF. Therefore, "first EOF1" would be redundant.

We replaced "EOF1" by "first EOF". Thanks

Line 43 Include a short sentence saying what the large-scale estimate by Neukom et al. (2014) suggests. Does it show a temperature anomaly in the MA during the LIA period?

We included in the main text: "This study suggests discrepancies in terms of timing and amplitude between air temperature variations in both hemispheres during LIA". Thanks

Line 95 "portion" suggests that you are doing a sub-diurnal analysis and that you compute the fraction of the total daily precipitation falling as snow. However, Eq. 2 suggests that you either consider all or none of the daily precipitation as snow.

We removed "portion" in the main text to clarify this point. Thanks

Line 106 The authors should say here that G(t) is not a variable available directly as a GCM model output, and it is instead estimated using the parametrization by Annandale et al. (2002), that is based on the minimum and maximum daily temperatures, as well as the relative position to the Sun. Refer to Appendix A1 for more details.

We added the following explanation into the main text: "Because solar radiation G(t) is not available as a GCM variable, we use a parametrization based on the daily minimum and maximum temperatures, as well as the relative position to the sun, following Annandale et al. (2002). Refer to Appendix A1 for more detailsÂĺ. Thanks.

Line 107 Although it is clear what you mean by snowpack here, the terminology is vague and not consistent throughout the manuscript. I would suggest referring to this consistently as firn.

We replaced "snowpack" by "firn". Thanks.

Line 107 You should add here that the actual albedo is calculated following Oerlemans and Knap (1998), a method that takes into account snow age and snow depth, and refer to Appendix A2 for details.

We added: "The surface albedo ($\alpha$) was calculated following Oerlemans and Knap (1998). This method takes into account snow age and snow depth. More details about this method can be found in Appendix A2". Thanks.

Line 110 Who obtained those factors? Pellicciotti? (be explicit). What do you mean by "based". Is that the average? How different are the values at Juncal Norte and San Francisco? Give a sense of the variability of those factors along the MA.

We included following clarification into our text: "These TF and SRF factor values are summarized on Ayala et al. (2017). They stem from in situ measurements in Juncal Norte glacier and were carried out by Pellicciotti et al. 2008. In the case of San Francisco, measurements are provided by Dirección General de Aguas DGA-MOP. Following Ayala et al. 2017, SRF on both glaciers takes similar values: 99 and 100 (mm h-1 W-1 m2) *10-4 in Juncal Norte and San Francisco glaciers, respectively. TF takes values of 100 and 50 (mm h-1 °C) *10-4 in Juncal Norte and San Francisco glaciers, respectively." Thanks.

Pellicciotti F, Helbing J, Rivera A, Favier V, Corripio J, Araos J, et al. A study of the energy balance and melt regime on Juncal Norte Glacier, semi-arid Andes of central Chile, using melt models of different complexity. Hydrol Process 2008;22:3980–97.

[Figure]

http://dx.doi.org/10.1002/hyp.7085.

Line 115 Eq. 3 say T mean > T crit , and here you say T mean > 0 . If the T crit of Eq.2 and Eq.3 refer to different parameters, please use a different symbol. Or just write T mean > 0 in Eq.3

We used "T mean > 0". Thanks

Line 125 "GCMs based on past1000 experiment simulations (runs r1i1p1) of the CMIP5 initia- tive". Explain what the "past1000" refers to, and give some detail of those CMIP5 runs. Otherwise, the wording in this sentence is confusing.

We replaced this sentence by a new paragraph, as follows: "We used daily climate data from three GCMs based on past1000 experiment simulations (runs r1i1p1) from the fifth phase of the Coupled Model Intercomparison Project initiative CMIP5 (Table 1). One of the aims of the past1000 project was to evaluate the natural variability in the climate system on centennial timescales. More details about the CMIP5 project are found in Taylor et al. 2012." Thanks.

Line 134 You refer to "this period" before actually defining it. Then define it in the following sentence. Please rephrase.

We modified the paragraph as follows: "In order to evaluate the capability of GCMs to reproduce the annual climatology of the MA region, we compare monthly precipitation and mean air temperatures from GCMs based on Historical CMIP5 simulations with measurements from the El Yeso meteorological station (YESO; 33°40 0 S; 70°05 0 W; 2475 m, no missing data). We compare both datasets over the 1979 – 2010 period". Thanks.

Line 135 You said in line 131 that "specific grid point information [of the GCM I guess] was used to compute temperature and precipitation lapse rates". However, here you say that you use a standard and constant lapse rate for temperature. And in line 139 you say that you also use a constant lapse rate for precipitation. Therefore, the statement in this line is misleading.

We clarified this point by removing this sentence. However, we included the following statement into the paragraph:

"Mean air temperature data were calculated for different elevations using a standard and constant lapse rate of -6.5 $^\circ$C * kmˆ$-1$. Due to the scarce number of studies about glacier-climate interactions in this part of the Andes, for minimum and maximum temperatures we used a constant lapse rate value of -5.5 $^\circ$C * kmˆ$-1$ following studies carried out in the Tropical Andes by Córdova et al. (2016). We used both temperatures to estimate Solar Radiation variable following Annandale et al. 2002, and described in section 2.1.1. In the case of precipitation, and given that the distribution of precipitation in mountainous regions is difficult to predict even under present-day conditions (Rowan et al., 2014), we use a constant rate of 0.02 mm * mˆ$-1$ in order to facilitate the computation of mass balance modelling and ELA estimation. In addition, elevation of each grid point by GCM were used to estimate temperature and precipitation lapse rates described before". Thanks

Line 136 You have not said why you need maximum and minimum temperature; they do not show up on any equations. This ambiguity will be solved if you introduce the suggestion made for line 106.

We resolved this point already. Thanks

Line 140 In general, there is no such thing as a "daily ELA" or "winter ELA" or "summer ELA", see general comments. Nevertheless, this daily and seasonal values seem irrelevant as they are not used anywhere in the results or discussion.

We remove "daily" and "seasonal" ELA designations from the text. Thanks

Line 149 Is it well established that the first EOF of the SST captures the ENSO signal? If so, give a reference at least.

We included this cite: Di Lorenzo, K. M. Cobb, J. C. Furtado, N. Schneider, B. T.

Anderson, A. Bracco, M. A. Alexander, and D. J. Vimont, 2010: Central Pacific El Niño and decadal climate change in the North Pacific Ocean. Nat. Geosci., 3, 762–765, doi:10.1038/ngeo984. Thanks.

Line 176 Again, it has not been explained why you need min/max temperatures to compute the ELA. See comments on line 136 and 106.

We explained this point before. Thanks.

Line 177 It says "period 1979 – 2015" but Figure A1 says 1979-2016.

We changed "1979-2016" by "1979 - 2015". Thanks

Line 186 How is the range of modeled ELA calculated? How do you estimate the uncertainty?

We included a new figure and a comparison between modeled ELA and ELAs retrieved from satellital imagery. Specifically, to assess our modeled annual ELA for the 1979 – 2015 period, we contrasted it with ELA information at annual resolution, obtained from Landsat images for five glaciers located across the Mediterranean Andes region (Juncal Norte, Olivares Gamma, Cipreses, Cortaderal and Universidad). Landsat images (MSS, TM, ETM+, OLI) have been widely used to obtain snowlines on glaciers (Rabatel et al., 2012, 2013; Wastlhuber et al., 2017; Rastner et al., 2019). The free access to images and the high acquisition frequency allows us to count with a long-time coverage of many glaciers worldwide. To contrast our modeled ELA with observations, we used ten years for comparison within the 1986 – 2014 period due to availability of images by this Andean region. In the results section, we comment that our ELA model presents similitudes with the ELA values from Landsat images (Figure 1). Our results show a good ELA representation for Juncal Norte and Olivares Gamma glacier (Figure 1a), which are located around of 33°00'S and 70°10'W within the MA region. Our modeled annual median ELA shows congruence with annual ELA values derived from Landsat images within 1979 – 2015. In this location of the MA region, our modeled ELA reproduces well the annual average ELA condition. In the case of satellite-derived ELAs from Universidad, Cortaderal, and Cipreses glaciers, our model shows an overestimation (Figure 1b). In the current version of the manuscript, we included a table that summarized the ELA values obtained by each glacier during the 1986 - 2014 years, and the total error obtained by each year.

Rabatel, A., Bermejo, A., Loarte, E., Soruco, A., Gomez, J., Leonardini, G., Vincent, C., and Sicart, J. E.: Can the snowline be used as an indicator of the equilibrium line and mass balance for glaciers in the outer tropics?, Journal of Glaciology, 58, 1027–1036, https://doi.org/10.3189/2012JoG12J027, 2012.

Rabatel, A., Francou, B., Soruco, A., Gomez, J., Cáceres, B., Ceballos, J. L., Basantes, R., Vuille, M., Sicart, J.-E., Huggel, C., Scheel, M., Lejeune, Y., Arnaud, Y., Collet, M., Condom, T., Consoli, G., Favier, V., Jomelli, V., Galarraga, R., Ginot, P., Maisincho, L., Mendoza,

J., Ménégoz, M., Ramirez, E., Ribstein, P., Suarez, W., Villacis, M., and Wagnon, P.: Current state of glaciers in the tropical Andes: a multi-century perspective on glacier evolution and climate change, The Cryosphere, 7, 81–102, https://doi.org/10.5194/tc-7-81-2013,https://www.the-cryosphere.net/7/81/2013/, 2013.

Rastner, P., Prinz, R., Notarnicola, C., Nicholson, L., Sailer, R., Schwaizer, G., and Paul, F.: On the Automated Mapping of Snow Cover on Glaciers and Calculation of Snow Line Altitudes from Multi-Temporal Landsat Data, Remote Sensing, 11, https://doi.org/10.3390/rs11121410, https://www.mdpi.com/2072-4292/11/12/1410, 2019.

Wastlhuber, R., Hock, R., Kienholz, C., and Braun, M.: Glacier Changes in the Susitna Basin, Alaska, USA, (1951–2015) using GIS and Remote Sensing Methods, Remote Sensing, 9, https://doi.org/10.3390/rs9050478, https://www.mdpi.com/2072-4292/9/5/478, 2017.

Line 192 This section is confusing, especially at this sentence. It should be more explicit that the authors are testing the ELA calculation method using a completely different dataset than the one used during the LIA period. Then, it is unclear the relevance of this comparison between the Carrasco ELA and the authors mean LIA ELA.

We included the next paragraph into the manuscript, in order to improve it: "At Universidad glacier, the previous ELA values reported by (Carrasco et al., 2005; Bravo et al., 2017; Kinnard et al., 2018) are within the range of the modeled ELA for the respective years (Figure A1b). In all cases, the annual observed ELA is within the range of the modeled ELA. However, in terms of absolute values, we found some discrepancies between ELA identified by Carrasco et al. (2005) and Kinnard et al. (2018) during 2000 and 2012, respectively. Carrasco et al. (2005) report a value of 3497 m.a.s.l. in 2000, while we obtained a modelled median ELA equal to 3722 m.a.s.l. On the other hand, Kinnard et al. (2018) report the ELA to be located at 3478 m a.s.l. in 2012, which is inconsistent with the median of 3980 m a.s.l. of our modelled ELA at Universidad glacier. On the other hand, despite to uncertainties of GCMs and based on the ELA equations reported by Carrasco et al. (2005, 2008), the present ELA is located at 4083 m.a.s.l for glaciers between 30° - 37°S, while our regional mean annual ELA during 1500 – 1848 CE was 3745 m.

Line 194 General comment to section 3.2: Is this the most appropriate way to compare/validate? Arguably, it would be more interesting to see how these models reproduce the inter- annual and decadal variability of YESO station.

We think that, given the nature of our aim research with focus on past climate variations, the reproduction of the annual climatology is an appropriate way to contrast GCMs and measurements such as YESO station in the MA region. We appreciate your comment. Thanks.

Line 196 "quite well" is too succinct and not substantiated. NCAR seems, and MRI seems to overestimate summer temperatures significantly. Also, Figure 2 excessively

aggre- gates the data. If we are looking for anomalies in a time series, it would be better to see the time series of El Yeso and the GCMs between 1979 and 2015, not just monthly means.

There is a correspondence in the aggregated data. We replaced "quite well" by "well". Thanks

Line 223 These "significant and common periodicities" were not obvious to me in figure 4. It would be interesting to highlight those period intervals in figure 4. Only the two year periods seem to be common to all models and maybe something around five years. A log-scale in the X-axis of figure 4 might help the visualization.

We appreciate your comment. Thanks. However, we think that wavelet analysis highlights and identifies well the spectral signals per se. Hence, it could be redundant to intervene further the spectrum. For clarification, significant signals were highlighted with black contours.

Line 240 What about the mismatch in timing of the periods with joint periodicities? The authors also talk about periods in-phase and anti-phase, which is quite confusing, and they do not address this later.

It is known that the relationship between large-scale climatic drivers and hydroclimatic variables in this region exhibits periods where it occurs synchronously (in phase), and other periods where the relationship is evident just after considering time lags (anti-phase, in the case that both phenomena are part of a succession). The ELA and SST time series considered in our work are non-stationary. The spectral analyses we used, i.e. cross wavelet and coherence, show periods where they share a strong spectral signal, but such relationship occurs after a certain lag. This is shown by vectors. We use the "phase" and "anti-phase" terminology only to refer to this process. We appreciate your comment. Thanks.

Line 261 Where was this comparison on the results? It does not seem to be there. Or

are the authors talking about the 1979-2010 comparison of figure 2? The latter is what line 263 seem to suggest.

Here, we refer to the 1979-2010 comparison of figure 2. Thanks.

Line 279 The authors can not know if there was a lower/higher ELA during the LIA compared with the second half of the past millennium because they only computed the ELA during the LIA. To know if there was a climate anomaly that could generate a glacier advance in the MA during the LIA, they need to compute the ELA over the whole past1000 data range, or at least in a range that extends beyond the LIA. With that information, they could see how the ELA during the LIA compares with the ELA before and after it, and with the prediction for the present (using past1000 runs data).

In this paragraph, we refer to the comparison with studies that were carried out in the northern hemisphere, which present information for 1500 - 1849 CE, the same period analyzed inour study. Please check the paragraph: Our modeled ELA in the MA region does not show longer intervals with a sustained low/high ELA (associated with positive/negative glacier mass balance), as identified in the Northern Hemisphere during the second half of the last millennium through lake sediments (Bakke et al., 2005), tree rings (Linderholm et al., 2007) and multiple climate proxies (Solomina et al., 2007). Thanks.

Line 292 Again, it would be nice to have those periods clearly highlighted in figure A4. Other- wise, it is difficult to see what the authors mentioned here.

We highlighted the periods. Please check Figure 2 attached. Thanks.

Line 313 It seems advisable to replace "values" with "anomalies" or "departures". Otherwise, the authors would be suggesting that there is such a thing as "negative precipitation".

We replaced "values" by "departures". Thanks.

Line 316 Where did the authors observed that? There is no such data in the results.

We removed this sentence. Thanks.

Line 329 Perhaps it would be better to say "might mask". If the authors say "seems to mask", they should explain better why it seems that way.

We replaced " seems to mask" by "might mask". Thanks.

Line 331 Consistent, but one would not expect to see such a quick response to climate. So better than saying "low ELA values around 1840" they could say it in the same way they do in the following paragraph: "low ELA between 1800-1848". The very low ELA they found around 1820 might have more to do with that maximum advance than the low ELA around 1840. Therefore, to associate the advance right away with the whole 1800-1848 period makes more sense.

We rephrased as follows: "The dominant influence of the Pacific SST variability seems to mask out the LIA signal that has been reported by many studies in the Northern Hemisphere (e.g., Luckman and Wilson, 2005; Solomina et al., 2015). Still, our results indicate low ELA values around 1840 (Figure 3) are consistent with a maximum advance in 1842 of Cipreses Glacier , located in the MA region, as documented by Araneda et al. (2009)". Thanks

Line 334 However, the authors can not say if it was colder than the centuries before or after that interval (1500-1848 CE), which is also very relevant for the discussion and the comparison with the northern hemisphere. A paragraph like this is missing in the conclusions.

Thanks for this comment. We plan to use the mass balance model presented here to reproduce ALS in the Andes and also in the Alps throughout the last millennium. There, we will analyze in depth, differences and similarities in terms of the ELA between the northern and the southern hemisphere.

Line 385 "R a for T t ". Perhaps "times" is a better term than "for" here.

We replace "times" by "for". Thanks

Line 387 T max should be T min

We replace "T max" by "T min". Thanks

Line 391 Why not use the elevation in the range 100-6000 m used for MB calculation? Please justify this choice.

We use the value of the elevation of each grid because we managed to acquire the topography considered by by each GCM. Otherwise, we planned to use the value extracted from the range used for the mass balance. We think that it might be useful to compare these approaches in future studies focused on estimations of modern ELA. We greatly appreciate this comment. Thanks.

Line 398 (Eq. A8) It would be better to use $\alpha$ ice instead of $\alpha$ hielo

We replace "hielo" by "ice" through text. Thanks.

Line 399 What does "global" mean in this context? Is it just the actual surface albedo?

Yes, indeed it is the actual surface albedo. We removed "global". Thanks.

Figure 5 Explain better how to interpret the arrows. The explanation in the caption is binary (in phase or anti-phase). However, the arrows can be seen in all directions. Do they display the angular phase difference?

We rephrased the caption as follows. Thanks. "Cross wavelet and wavelet coherence between each regional annual ELA and EOF1 from SST over the Niño 3.4 region (April to March) from the respective GCM. The (a), (c) and (e) panels represent the cross wavelet, while (b), (d) and (f) show the wavelet coherence analysis. The arrows within black contours of each panel indicate that the two time series vary in-phase or anti-phase. A zero phase difference means that both time series evolve synchronously. Arrows point to the right (left) when the time series are in phase (anti-phase). "In phase" ("anti-phase") indicates that both time series are positively (negatively) correlated. The thick black contours indicate the significance level at P < 0.05 obtained from a red noise

model. Areas outside the cone of influence are shown in a lighter shade."

Yes, arrows represent the angular phase differences between both time series. We recommended learn this free access guide about wavelet. http://www.hs-stat.com/WaveletComp/weather_and_radiation.php

Figure 6 It would be nice to have a box showing the El Niño 3.4 region as well as the MA.

Done. Thanks for your suggestion.

Figure A1 What the do the bar sizes, error bars and green circles mean?

We replaced figure A1. Please check Figure 3 attached. The caption is as follows: "Modeled equilibrium line altitude (ELA) and inferred ELAs from Landsat imagery of a) Juncal Norte and Olivares Gamma glaciers, and b) Cortaderal, Cipreses and Universidad glaciers for the period1979 - 201. Circles indicate the specific annual ELA by each glacier located in the Mediterranean Andes region. Vertical lines indicate the total error. The glaciological model has been forced by a gridded product of daily precipitation, and mean, minimum and maximum daily temperatures obtained from Chilean meteorological stations. The horizontal resolution is 5-km, and the data has been compiled by Climate and Resilience center (CR)2. The data can be downloaded freely from: http://www.cr2.cl/datos-productos-grillados/?cp_cr2met=2

Technical corrections

We added directly your corrections and suggestions in our new version of the manuscript. Many thanks

––––––––––––––––––––––––––––––

[Figure]

**Fig. 1.**

**Fig. 2.**

**Fig. 3.**